

# A new species of *Ampharete* Malmgren, 1866 (Annelida: Ampharetidae) from Washington and redescription of *A. cirrata* Webster & Benedict, 1887 and *A. labrops* Hartman, 1961

Yessica Chávez-López

Departamento de Sistemática y Ecología Acuática, El Colegio de la Frontera Sur, Chetumal, Quintana Roo, Mexico

## ABSTRACT

*Ampharete acutifrons* (Grube, 1860), originally described from Greenland, has long been considered a widely distributed arctic-boreal species. However, recent morphological re-assessment of the holotype indicates that most previous records of *A. acutifrons* were misidentifications, and molecular sequence data also suggest that *A. acutifrons* is a multispecies complex. This study focuses on specimens of the *A. acutifrons* species complex from Washington, USA, with publicly available cytochrome c oxidase subunit I (COI) sequence data. Specimens from Washington belonging to the Invertebrate Zoology Collection of the Florida Museum of Natural History were examined. Additional specimens were examined for morphological comparison, including type material of *A. cirrata* Webster & Benedict, 1887, and *A. labrops* Hartman, 1961. Detailed morphological descriptions of specimens and photographs of the diagnostic characters were made. The molecular analysis includes 37 published COI sequences of *Ampharete* and *Anobothrus* species sourced from public databases. Redescriptions of type material of *A. cirrata* and *A. labrops* are provided. *Ampharete paulayi* **n. sp.** is described as a new species from Washington, USA, based on morphological and COI sequences data. Photographs of living specimens are presented, a hypothesis on the development of buccal tentacles in *Ampharete* species is proposed, and the use of Methyl green stain is recommended as a standard practice in future descriptions of ampharetids.

## INTRODUCTION

The family Ampharetidae *Malmgren, 1866* comprises small, tubicolous benthic marine annelids commonly found in soft sediments from intertidal zones to deep sea environments (*Rouse, Pleijel & Tilic, 2022*). Within Ampharetidae, the genus *Ampharete Malmgren, 1866* is the most species-rich, currently comprising about 50 nominal species and six synonymized genera (*Kim, Kim & Jeong, 2025*; *Read & Fauchald, 2025*).

Corresponding author
Yessica Chávez-López,
yess.chl05@gmail.com

For a long time, *Ampharete acutifrons* (*Grube, 1860*) was considered the type species of *Ampharete*, although this designation is currently considered incorrect (Read 2022 in *Read & Fauchald, 2025*). Particularly, *A. acutifrons* was described based on a single specimen from Greenland (*Grube, 1860*). Historically, *A. acutifrons* has been considered a broadly distributed arctic-boreal species, characterized by the presence of long dorsal cirri on posterior thoracic neuropodia and all abdominal neuropodia (*e.g.*, *Hessle, 1917*; *Annenkova, 1929*; *Jirkov, 2001*). In addition, *A. grubei Malmgren, 1866* and *A. cirrata Webster & Benedict, 1887* have been considered as junior synonyms of *A. acutifrons* (*Hessle, 1917*).

Recent re-examination and redescription of the holotype of *A. acutifrons* revealed that the neuropodial cirri are absent in the type (*Krüger et al., 2022*). Unfortunately, the holotype was noted to be in poor condition, strongly contracted and without branchiae in the original description (*Grube, 1860*), and no other specimens from Greenland have been described for comparison. Despite this, re-examination of type material allows the reestablishment of *A. grubei* and *A. cirrata* as valid species, which differ from *A. acutifrons* in the shape of the paleae and the presence and size of the neuropodial cirri (*Krüger et al., 2022*). Based on this morphological information, many prior records of *A. acutifrons* are likely to be misidentifications. This idea is also supported by molecular data, which suggest that *A. acutifrons* is a species complex (*Krüger et al., 2022*), and includes some specimens from Washington, USA.

In light of these recent discoveries, previous records of *A. acutifrons* are in need of re-evaluation. This study focuses on *Ampharete* specimens from Washington with associated cytochrome c oxidase subunit I (COI) barcode sequences available in public databases. Detailed morphological examination of museum specimens and analysis of their COI sequences reveal three *Ampharete* species in Washington, including a new species formally described here: *A. paulayi* **n. sp.**

## MATERIALS AND METHODS

The studied material belongs to the following collections: Invertebrate Zoology Collection of the Florida Museum of Natural History (UF), University of Florida, USA; Natural History Museum (BMNH), London, UK; Polychaete Collection of the Natural History Museum (LACM-AHF), Los Angeles County, USA; Swedish Museum of Natural History (SMNH), Stockholm, Sweden.

Species descriptions are based solely on external characters observed in holotype specimens. Additional morphological information from paratypes or non-type specimens is included in a separate "Variation" section. The material examined citations follow the standardized format of *Chester et al. (2019)*.

In *Ampharete*, the prostomium is fused ventro-laterally with the peristomium to form a structure commonly referred to as the lower lip. As the prostomium and peristomium (including lower lip) are pre-segmental regions (*Rouse, Pleijel & Tilic, 2022*), they are excluded from the body segment counts in this study. The arrangement and number of branchiae are commonly used to distinguish species (*Reuscher, Fiege & Wehe, 2009*); however, their origin and subsequent enumeration often vary among descriptions due to

the lack of standardization. Therefore, this study considers that there are only one pair of branchiae per segment (*Orrhage, 2001*); the first thoracic segment is achaetous and has no branchiae; and the second segment bears the first pair of branchiae and is the first thoracic chaetiger, as it has paleae. To avoid confusion, segments are indicated by Roman numerals, while chaetigers are indicated by Arabic numerals (*Reuscher, Fiege & Wehe, 2009*). I follow *Jirkov (2009)* regarding the shape of the prostomium of *Ampharete*, subdivided by a U-shaped groove that separates it into a middle lobe and a surrounding lobe, giving the prostomium a trilobed appearance at the anterior margin. The terms "tori" and "pinnules" refer to the shape of the neuropodia: low and simple ridge-like when they are tori, and wide and fan-shaped when they are pinnules (*Holthe, 1986b*). The term "intermediate uncinigers" is used to refer to the tori form of abdominal neuropodia (*Imajima, Reuscher & Fiege, 2012*). Following *Chávez-López, Alvestad & Moore (2025)*, the intermediate uncinigers are included in the abdominal segment count.

Some specimens were temporarily stained with Shirlastain-A or Methyl green to improve the visibility of nephridial papillae, the branchial arrangement and neuropodial cirri (see 'Morphology' section). One thoracic and abdominal unciniger (generally TU4 and AU4) were dissected to describe uncini morphology. Permanent slides of tentacles, paleae and uncinigers were prepared using Euparal mounting medium and incorporated into the corresponding collection. Digital photographs of complete specimens and diagnostic characters were made using a Canon T8i digital camera equipped with a microscope adapter. Series of digital photographs were combined through focus stacking (Z-stacking) using Helicon Focus v.8.2.15 software (*Helicon Soft Limited, 2025*). Figures were prepared using Adobe Photoshop v.25.12.3 software (*Adobe Inc, 2024*).

The following abbreviations are used in the text and figures: abdominal unciniger (AU), branchiae originating from segment II (Br1), branchiae from segment III (Br2), branchiae from segment IV (Br3), branchiae from segment V (Br4), lateral cirri of pygidium (Lc), thoracic chaetiger (TC), thoracic unciniger (TU), peristomium (Pe).

### Molecular analysis

Thirty-seven COI sequences published in the NCBI GenBank and Barcode of Life Data System (BOLD) databases were used for molecular analysis (Table S1). All molecular analyses were performed in MEGA version 11 (*Tamura, Stecher & Kumar, 2021*). COI sequences were aligned using MUSCLE (*Edgar, 2004*). Model testing was performed using maximum likelihood; the best-fitting substitution model was selected according to the Bayesian Information Criterion (BIC). A maximum likelihood tree was reconstructed applying the General Time Reversible (GTR) model using a discrete Gamma distribution (+G) with five rate categories and by assuming invariable sites (+I). Model-corrected inter- and intraspecific genetic distances were estimated using Kimura's two-parameter (K2P; *Kimura, 1980*) distance model.

### Nomenclatural acts

The electronic version of this article in Portable Document Format (PDF) will represent a published work according to the International Commission on Zoological Nomenclature

(ICZN), and hence the new names contained in the electronic version are effectively published under that Code from the electronic edition alone. This published work and the nomenclatural acts it contains have been registered in ZooBank, the online registration system for the ICZN. The ZooBank LSIDs (Life Science Identifiers) can be resolved and the associated information viewed through any standard web browser by appending the LSID to the prefix http://zoobank.org/. The LSID for this publication is: urn:lsid:zoobank.org:pub:F771A2B4-4F5F-4168-BBAD-A519175C83AE. The online version of this work is archived and available from the following digital repositories: PeerJ, PubMed Central SCIE and CLOCKSS.

## RESULTS

### Molecular comparison

The COI barcode sequences for some specimens examined here were previously deposited in BOLD and identified as *Ampharete acutifrons*. The phylogenetic analysis of COI recovered three distinct genetic lineages, herein identified as: *A. cirrata Webster & Benedict, 1887*, *A. labrops Hartman, 1961*, and *A. paulayi* **n. sp.** (Fig. 1). *Ampharete labrops* from Washington were closely related to specimens identified as *A. labrops* from Vancouver Island, with a K2P-corrected genetic distance of 1.3–1.7%.

The intraspecific K2P genetic distance variation was low within the *A. cirrata* population from Washington (0.2–0.5%). The K2P-corrected genetic distance between the Baltic Sea and Washington populations is 4.4–4.9% (Table S2), yet few morphological differences distinguish these populations and the syntype of *A. cirrata* (see "*A. cirrata* Variation"). This suggests a recent history of genetic isolation in these populations and may reflect and ongoing, incomplete speciation process.

All included *Ampharete* species had K2P-corrected COI distances between 15.4–22% for *A. paulayi* **n. sp.** (Tables S2, S3), a result consistent with the morphological analysis. The sequences within *A. paulayi* **n. sp.** were identical. This result supports the hypothesis that paratypes of *A. paulayi* are juveniles with an incomplete development of some structures, such as the buccal tentacles or abdominal pinnules (see "*A. paulayi* **n. sp.** Remarks"). At the same time, it confirms earlier accounts of ontogenetic changes in some *Ampharete* species.

### Morphology

The shape of thoracic and abdominal neuropodia has long been considered an important character for distinguishing ampharetid species. In *Ampharete*, neuropodia may bear dorsal marginal cirri or papillae. In other annelids, the term 'cirri' generally refers to dorsal or ventral appendages of the prostomium or parapodia (*e.g.*, *Rouse, Pleijel & Tilic, 2022*), rather than to the overall shape of the parapodial ramus; in the latter context, the term 'cirriform' is more appropriate. In contrast, 'papillae' traditionally refers to epidermal rugosities containing glandular cells and pores, which may have secretory, sensory or combined functions (*e.g.*, *Fauchald & Rouse, 1997*; *Vodopyanov, Tzetlin & Zhadan, 2014*). In ampharetids, however, there is no evidence that the 'neuropodial papillae' are secretory.

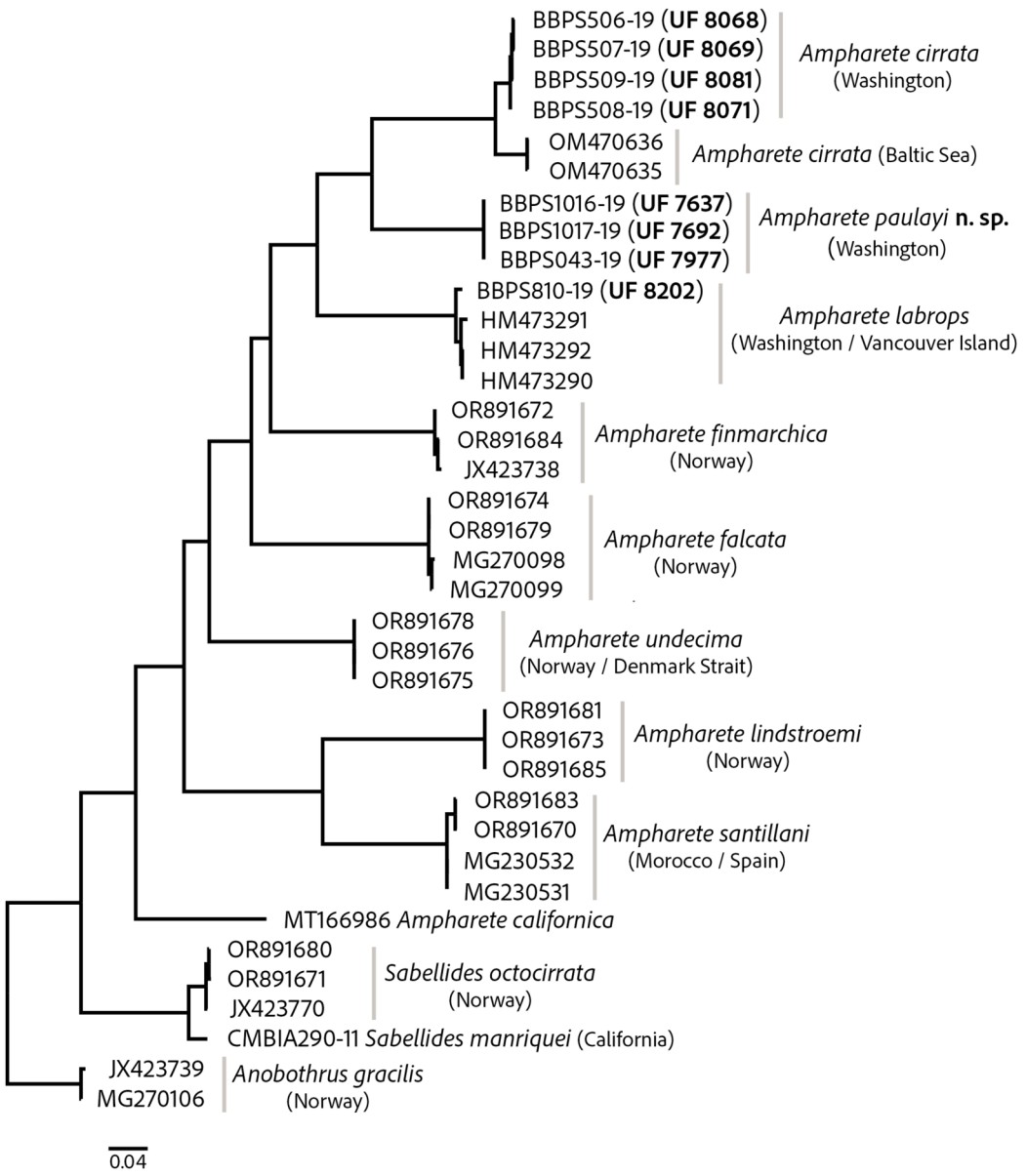

**Figure 1 Maximum likelihood (ML) tree of COI sequences.** Molecular analysis based on GTR model using a discrete Gamma distribution with five rate categories and by assuming that a certain fraction of sites is evolutionarily invariable. Labelling next to the species names indicate the GenBank or BOLD accession numbers.

Instead, the distinction between neuropodial papillae and cirri lies only in their shape: cirri are elongated and pointed, whereas papillae are small and rounded. Therefore, referring to the same structure by different names depending on its shape is problematic and confusing. Moreover, in some species (*e.g.*, *A. cirrata*), the so-called 'neuropodial papillae' of anterior segments gradually transform into 'neuropodial cirri' in posterior segments. Based on these observations, the terms 'neuropodial cirrus' (singular) and 'neuropodial

cirri' (plural) were chosen in this study as the most appropriate to refer to the dorsal projection of the neuropodia, which can be rounded or elongated.

## Systematics

**Family Ampharetidae** *Malmgren, 1866*
**Subfamily Ampharetinae** *Malmgren, 1866*

***Ampharete** Malmgren, 1866*

**Type species.** *Amphicteis acutifrons Grube, 1860*, by subsequent designation (*Uschakov, 1955*: 366).

**Diagnosis.** Ampharetins with 17 thoracic segments: 15 chaetigers (including paleal segment), 12 uncinigers from segment VI (TC4). Prostomium subdivided into two lobes, without glandular ridges. Segment II with paleae. Four pairs of branchiae; three pairs of branchiae arranged in a transverse row with an interbranchial gap, fourth pair immediately behind them. Two dorsal nephridial papillae next to fourth pair of branchiae, well separated from each other. Buccal tentacles pinnate. Pygidium with two lateral cirri, some species also with several surrounding anal cirri.

**Remarks.** *Ampharete acutifrons* (*Grube, 1860*) was subsequently designated as the type species of *Ampharete* by *Uschakov (1955*: 366), although several authors (*e.g.*, *Holthe, 1986a*; *Krüger et al., 2022*) have mistakenly attributed it to *Hartman (1959)*. The validity of *A. acutifrons* as the type species is currently in doubt, mainly because it was not among the nominal species originally included when the genus *Ampharete* was established. Read (2022 in *Read & Fauchald, 2025*) has suggested that *A. grubei Malmgren, 1866* should instead be considered the type species of *Ampharete*. This informal proposed was probably followed by *Jirkov (2023*: 3), who considered *A. grubei* as the type species of *Ampharete*. However, this is a nomenclatural issue that requires formal consideration by the International Commission on Zoological Nomenclature (ICZN). Pending an official resolution, this study continues to consider *A. acutifrons* as the type species of *Ampharete*.

Several genera have historically been treated as junior synonyms of *Ampharete Malmgren, 1866*: *Asabellides Annenkova, 1929*, *Branchiosabella Claparède, 1863*, *Heterobranchus Wagner, 1885*, *Pseudosabellides Berkeley & Berkeley, 1943*, *Pterampharete Augener, 1918* and *Sabellides* Milne-Edwards in *Deshayes & Milne-Edwards, 1838* (*Fauvel, 1897*; *Day, 1964*; *Jirkov, 1994*, *1997*, *2018*). However, I do not follow the proposed synonymies of *Asabellides*, *Pseudosabellides*, *Pterampharete* and *Sabellides* with *Ampharete*. The diagnostic characters of these genera are inconsistent with *Malmgren's (1866)* definition of *Ampharete*, particularly regarding the presence of paleae (*i.e.*, chaetae often longer and wider than typical capillary notochaetae) on segment II, and the presence of 12 thoracic uncinigers. Accordingly, the diagnosis of *Ampharete* provided in this study is restricted to adult specimens, and excludes the characters attributed to species of *Asabellides*, *Pseudosabellides*, *Pterampharete* and *Sabellides*.

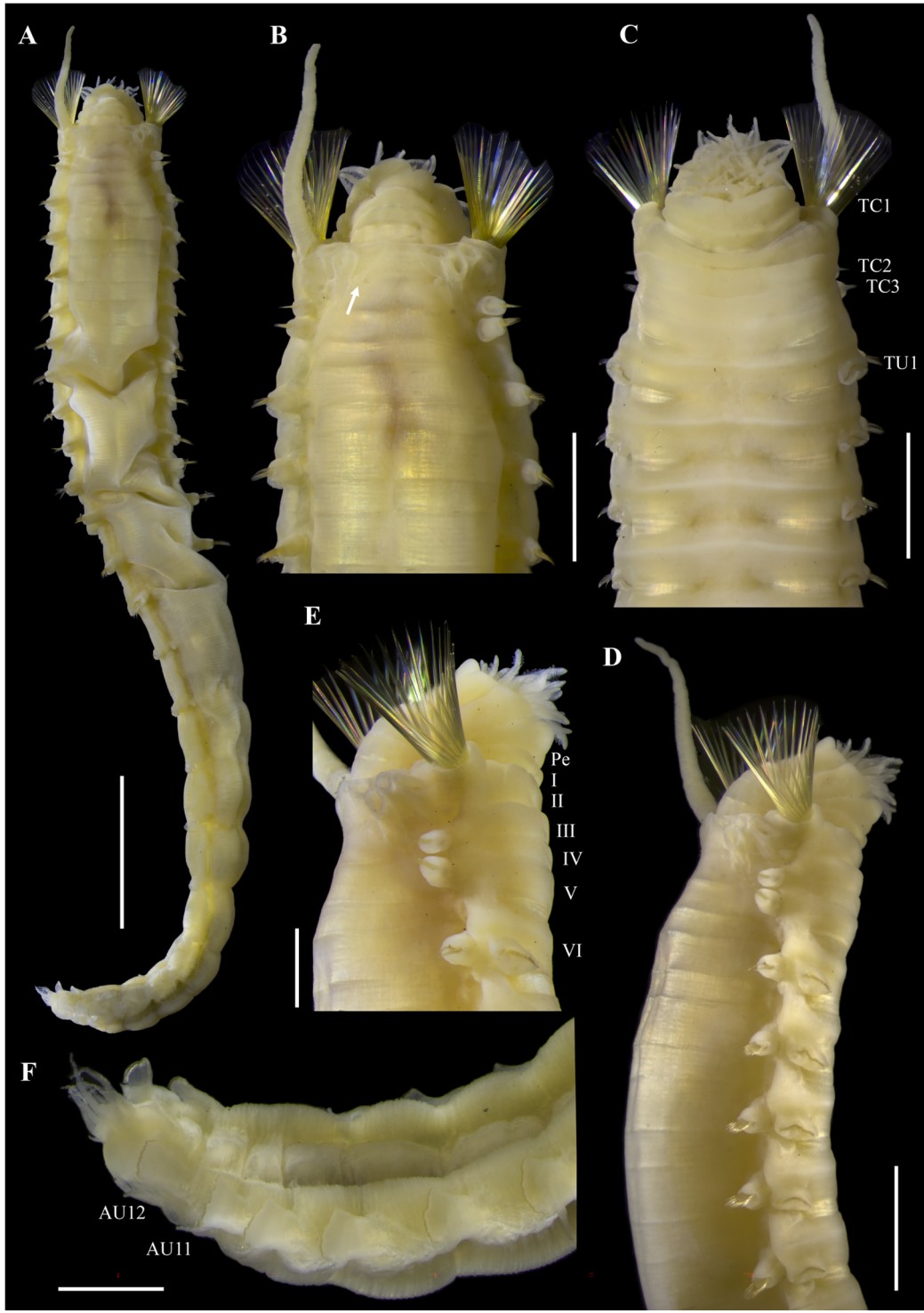

**Figure 2** *Ampharete paulayi* n. sp., holotype (UF 7977). (A) Complete specimen, dorsal view. (B) Anterior region, dorsal view (arrow points dorsal nephridial papilla). (C) Same, ventral view. (D) Same, lateral view. (E) Close-up of anterior thoracic chaetigers, lateral view. (F) Posterior region, lateral view. Scale bars: A: 5 mm; B–D: 2 mm; E–F: 1 mm.     

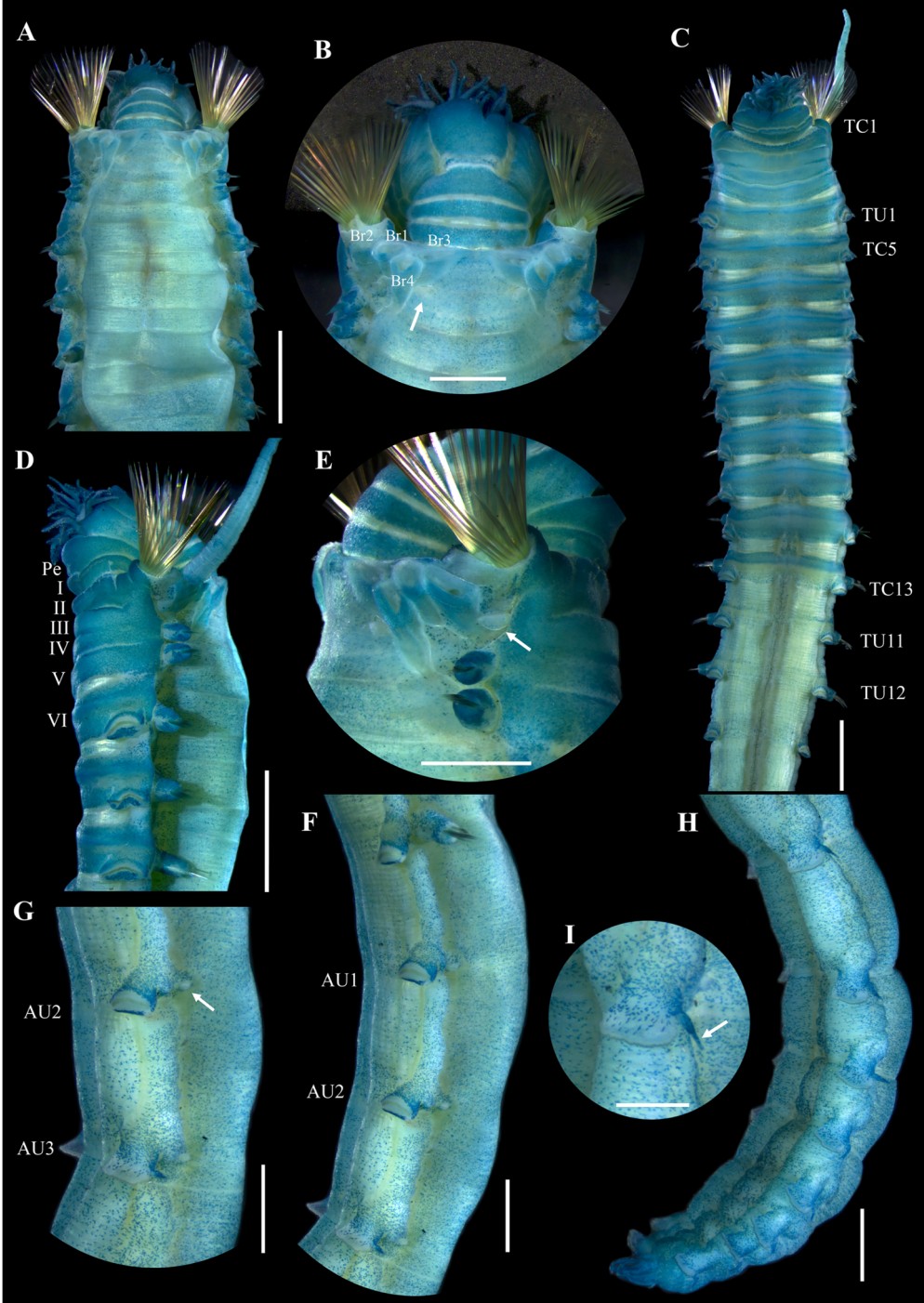

**Figure 3** *Ampharete paulayi* **n. sp., holotype (UF 7977) after Methyl green staining.** (A) Anterior region, dorsal view. (B) Close-up of prostomium and branchial arrangement, dorsal view (arrow points dorsal nephridial papilla). (C) Thorax, ventral view. (D) Anterior region, lateral view. (E) Close-up of anterior chaetigers, lateral view (arrow points rudimentary notopodium on segment III). (F) First abdominal segments, lateral view. (G) Intermediate uncinigers (AU2, arrow points rounded dorsal tubercle) and abdominal pinnules (AU3). (H) Posterior abdominal segments, lateral view. (I) Close-up of abdominal pinnules (arrow points the stained neuropodial dorsal). Scale bars: A, C–D: 2 mm; B, E–H: 1 mm; I: 0.5 mm.

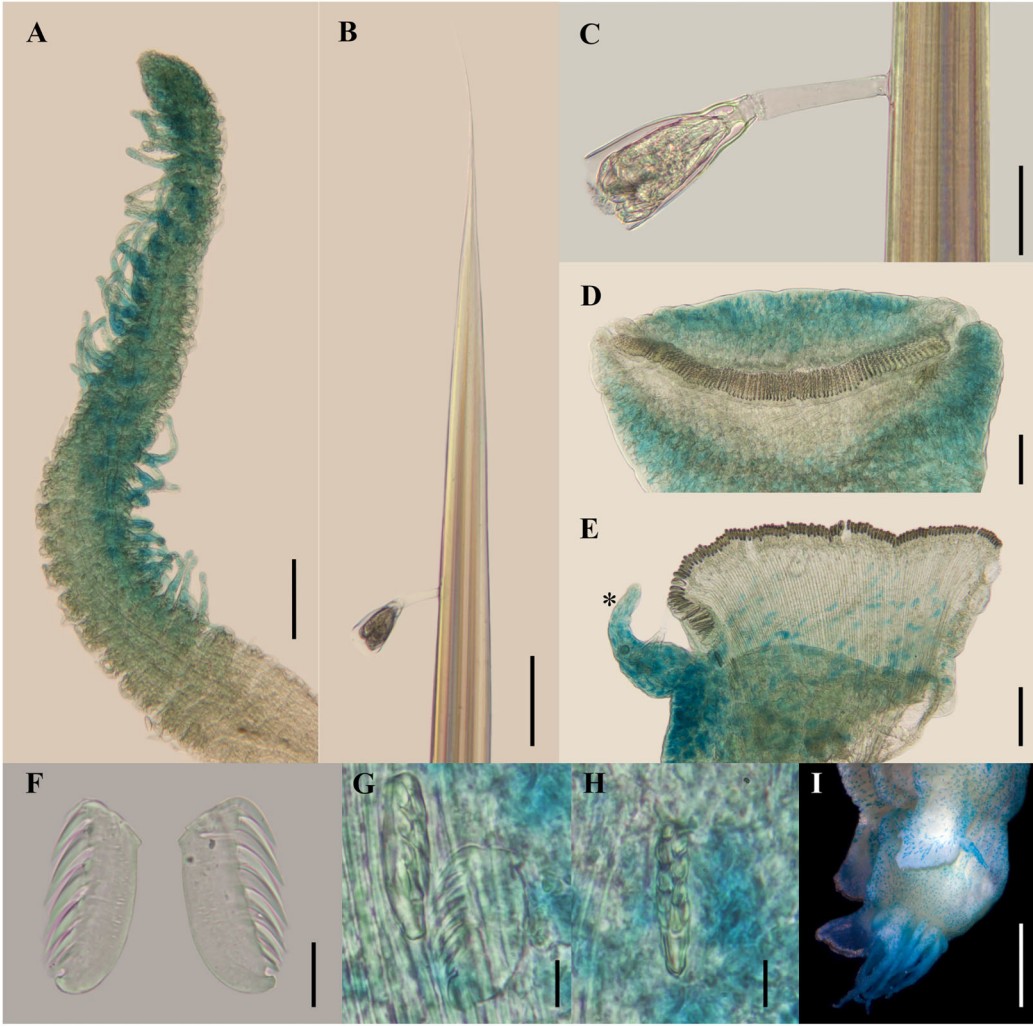

**Figure 4** *Ampharete paulayi* **n. sp., holotype (UF 7977).** (A, D–E, G–I) After Methyl green staining. (A) Buccal tentacle. (B) Palea. (C) Close-up of ciliate attached to palea. (D) Thoracic unciniger 4 (seg. IX). (E) Abdominal pinnules AU4 (asterisk marks the stained neuropodial cirrus). (F) Thoracic uncini, lateral view. (G) Abdominal uncini, lateral view. (H) Same, frontal view. (I) Pygidial cirri. Scale bars: A–B, D–E: 100 μm; C: 40 μm; F: 20 μm; G–H: 10 μm; I: 0.5 mm.

*Ampharete paulayi* **n. sp.**

Figures 2–5

**Diagnosis**. *Ampharete* with 26–28 paleal chaetae per side on segment II. All thoracic neuropodia entire, without cirri. Abdomen with 12 segments. Each abdominal pinnules (AU3–12) with an elongate neuropodial cirrus, progressively increasing in length toward posterior segments. Pygidium with two lateral cirri and 14 long cirri surrounding the anus.

**Etymology**. The species is name in honor of Dr. Gustav Paulay, curator of the Invertebrate Zoology collection at UF, who collected most of the specimens reviewed in this work and who supported the examination of material in the collection under his care. The species

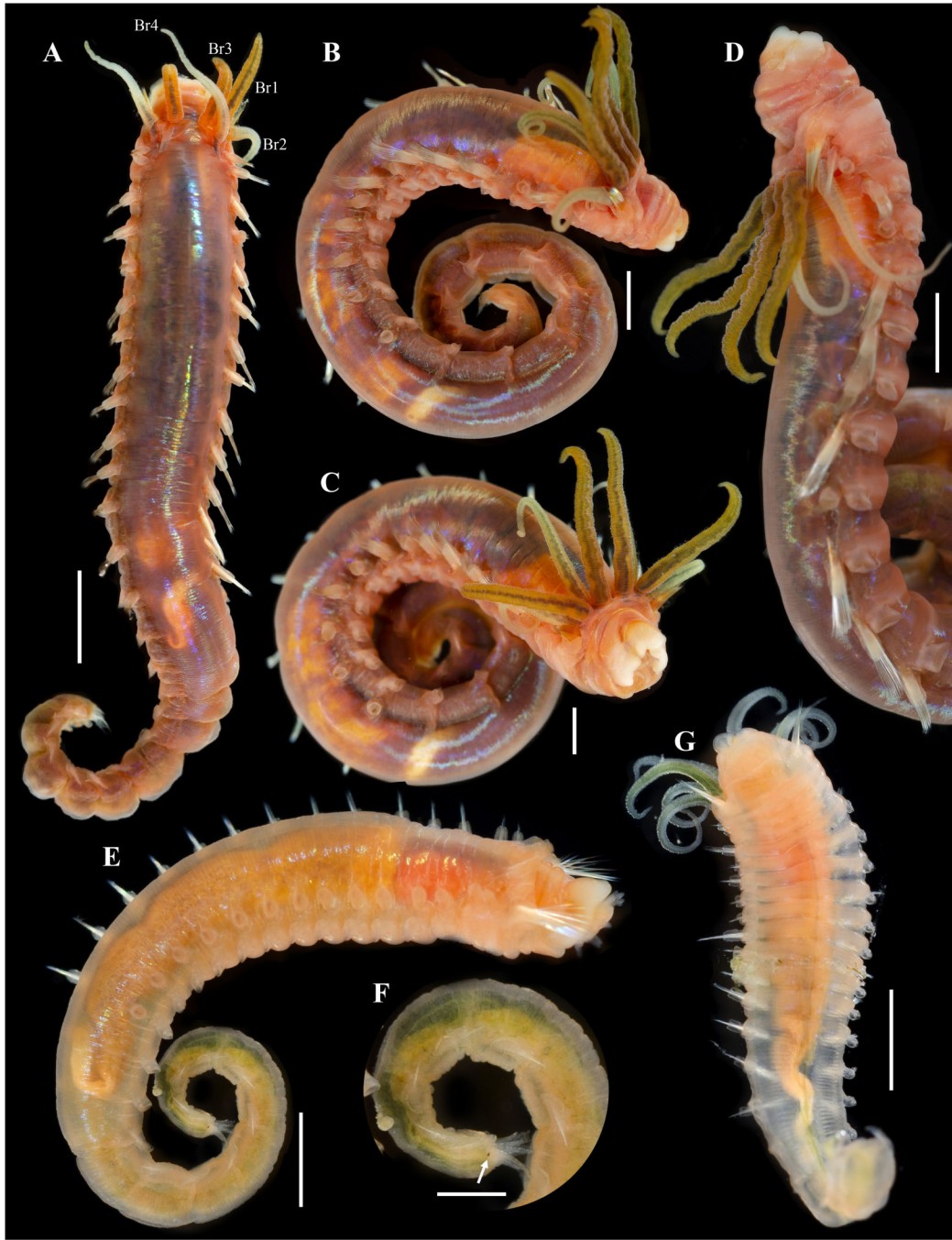

**Figure 5** *Ampharete paulayi* **n. sp., types in life.** (A–D) Holotype (UF 7977). (A) Complete specimens, dorsal view. (B–C) Same, dorso-lateral view. (D) Same, lateral view. (E, F) Paratype (UF 7637). (E) Complete specimen, dorso-lateral view. (F) Close-up of pygidium with eyespots. (G) Paratype (**UF 7692**), ventral view. Scale bars: A: 3 mm, B–D: 2 mm; E: 0.5 mm; F: 0.2 mm; G: 1 mm. Photos credit: Gustav Paulay.
name is a noun in the genitive case (*ICZN (International Commission of Zoological Nomenclature), 1999*, Art. 31.1.2).

**Material examined**

**Holotype**

WASHINGTON • 1 complete spec.; Kitsap County, North Hood Canal, south of the bridge; 47.83758°N, 122.62895°W; depth 21 m; 17 Apr. 2019; Gustav Paulay leg.; soft bottom; BOLD COI: BBPS043-19 (identified as *A. acutifrons*); **UF 7977**.

**Paratypes**

WASHINGTON • 1 complete spec.; Pierce County, west of Devils Head, eastern end of Nisqually Reach; 47.17060°N, 122.78051°W; 10 Apr. 2019; depth 109.25 m; Gustav Paulay leg.; soft bottom; BOLD COI: BBPS1016-19 (identified as *Ampharete*); **UF 7637** • 1 incomplete spec.; same data as previous; BOLD COI: BBPS1017-19 (identified as *Ampharete*); **UF 7692**.

**Description.** Holotype complete (**UF 7977**), body pale-yellow, 33 mm long, 5 mm wide (Fig. 2A). Prostomium subdivided into two lobes (Figs. 2B, 3A, 3B), middle lobe anteriorly rounded, one pair of nuchal organs as ciliated pits (Fig. 3B), eyespots not observed. Buccal tentacles partially everted (Figs. 2C, 2D), bipinnate (Fig. 4A). Lower lip entire (Figs. 2C, 3C).

Thorax with 17 segments: 15 chaetigers, 12 uncinigers. Segment I achaetous, clearly distinguishable in ventral view, not fused with peristomium (Figs. 2E, 3D). Segment II with paleae arranged in spiral (Figs. 3B, 3E): 28 chaetae on the right side, 26 chaetae on the left. Paleal chaetae yellowish, thick and flat with caudate tips (Fig. 4B). Paleal chaetae tips very fragile, most paleae with broken tips. Paleae 1–1.8 mm long; longest paleae protruding beyond prostomium (Figs. 2B–2D), four times longer than TC2 notochaetae (0.4 mm), and thrice as long as the longest notochaetae (TC8, 0.6 mm). Some ciliate epibionts attached to paleae (Fig. 4C).

Four pairs of branchiae originating from segments II–V (Figs. 2B, 3B). Branchiae arranged in two transverse rows. Three pairs of branchiae arranged in a transverse row on segments II–III, separated by a median gap as long as branchial width; 4th pair shifted behind innermost (Br3) and middle (Br1) branchiae of transverse row with bases extended to segment V (TC3) (Fig. 3B). Most branchiae lost, only visible as scars, except outermost branchiae (Br2), left side (Figs. 2A–2D, 3C, 3D). Branchia filiform, smooth, 3.5 mm long, reaching back to thoracic chaetiger 5 (TU2); branchia detached by handling, but present in vial. Two dorsal nephridial papillae between 4th pair of branchiae (Figs. 2B, 3B). Nephridial papillae well separated by a wide gap of 1 mm, ~7 times as long as nephridial papillae width (Fig. 3B).

Notopodia with capillary chaetae on 14 segments from segment IV (Fig. 2E). Segment III achaetous, fused dorsally with segment II, easily distinguishable on ventral and lateral views (Figs. 2E, 3D), with rudimentary notopodia (Fig. 3E). First two thoracic notopodia slightly displaced dorsally (TC2–3) (Fig. 3D). All notopodia lobes of similar size, as long as wide. Notochaetae arranged in two rows, each with 8–10 capillary bilimbate chaetae. Nephridial papillae below notopodial lobes of TC4 (TU1).

Neuropodia tori present on 12 segments from chaetiger 4 (segment VI) (Fig. 3C). Thoracic neuropodia smooth, without cirri (Fig. 4D). Thoracic uncini with two rows of denticles in front view, 7–8 denticles in lateral view (Fig. 4F). Ventral shields along chaetigers 1–13 (segments II–XV) (Fig. 3C). Continuous median ventral groove visible from chaetiger 14 (segment XVI, TU11) to posterior end.

Abdomen with 12 chaetigers (Fig. 2A). Small and rounded dorsal tubercle (rudimentary notopodia?) along abdominal segments 1–9 (Figs. 3F, 3G). First two segments with intermediate uncinigers (Fig. 2G). Third and following abdominal segments with pinnules. Abdominal pinnules with elongate neuropodial cirri (Figs. 3H, 3I). Anterior pinnules with shorter cirri, as long as pinnules length (Figs. 3I, 4E). Neuropodial cirri progressively longer towards posterior segments, slightly exceeding pinnules length (Figs. 2F, 4I). Abdominal uncini with two rows of denticles in frontal view, 6–7 denticles in lateral view (Figs. 4G, 4H).

Pygidium with two long broad lateral cirri, one bifurcated, and about 14 long cirri surrounding anus in one row (Figs. 2F, 4I). Pygidial cirri elongate, 0.3–0.9 mm long, longest cirri ~11 times longer than wide, reaching forward to last abdominal segment.

**Methyl green staining pattern.** Prostomium intensely stained, except for margins of middle lobe and nuchal organs (Figs. 3A, 3B). Peristomium and segment I completely stained, except for intersegmental areas (Figs. 3C, 3D). Buccal tentacles completely stained, except basally lacking lateral filaments; base of tentacular filaments strongly stained (Fig. 3A). Thorax dorsally spotted (Fig. 3A). Thoracic notopodia intensely stained, except laterally (Figs. 3D, 3E). Thoracic neuropodia intensely stained around uncini row (Figs. 3D, 4D). Ventral shields strongly stained from TC1–13 (Fig. 3C). Abdomen spotted (Figs. 3F–3H). Abdominal pinnules with neuropodial cirri intensely stained (Figs. 3I, 4E). All anal cirri intensely stained (Fig. 4I).

**Body coloration in life.** Holotype with body pinkish, dorsally iridescent (Fig. 5A). Prostomium pink, with anterior margin whitish (Figs. 5B–5D). Branchiae two-toned: innermost and middle pairs greenish-orange, outermost pair and 4th pair pale; all branchiae with dark blood vessels visible by transparency (Figs. 5A–5D). Paleal chaetae translucent yellowish. Notochaetae translucent white. Pygidial cirri pale (Fig. 5A).

Paratypes with translucent orange to melon-colored body (Figs. 5E–5G). Paratype **UF 7637** with a pygidial eyespot (Figs. 5E, 5F). Paratype **UF 7692** with branchiae translucent green to pale (Fig. 5G).

**Variations.** The type series of *A. paulayi* **n. sp.** consisted of three specimens: complete holotype and two paratypes, one complete (**UF 7637**) and another one incomplete (**UF 7692**). Complete specimens 15–33 mm long, 1.7–5 mm wide, 14–28 paleal chaetae per side (0.8–1.8 mm long), 12 abdominal segments, and 10–17 long anal cirri. The complete paratype (15 mm long, 1.7 mm wide) has short smooth tentacles, and abdominal pinnules with rounded cirri instead of bipinnate tentacles and abdominal neuropodia with elongate cirri. The incomplete paratype (5 mm long, 1.2 mm wide), with only one abdominal segment, also has short smooth tentacles, but there is a single larger bipinnate tentacle.

**Remarks.** *Ampharete paulayi* **n. sp.** has 15 thoracic chaetigers, 12 thoracic uncinigers, paleae on segment II, rudimentary notopodia without chaetae on segment III, 12 abdominal segments, abdominal pinnules with elongate neuropodial cirri (as long as pinnules length), and 10–17 long cirri surrounding the anus.

Paratypes of *A. paulayi* have short, smooth tentacles and abdominal pinnules with rounded neuropodial cirri. The presence of smooth tentacles differs with the holotype and is even inconsistent with the diagnosis of *Ampharete*. However, considering the size of the specimens, it is likely that the holotype (33 mm long) is an adult specimen, whereas the paratypes (<15 mm long) are juveniles of *A. paulayi* **n. sp.** In the larval development of *A. acutifrons* (*e.g.*, *Clavier, 1984*), *A. grubei* (*e.g.*, *Fauvel, 1897*) and *A. labrops* (*Blake, 2017*), the only three *Ampharete* species studied to date, the buccal tentacles are smooth and ciliated in larval stages but become bipinnate in adults. The presence of one larger bipinnate tentacle in the paratype **UF 7692** of *A. paulayi* **n. sp.** reinforces the hypothesis that pinnate tentacles develop in later juvenile stages. The same may be true for the development of neuropodial cirri in the abdominal segments.

There are no differences in other diagnostic characters nor in the COI sequences of the *A. paulayi* **n. sp.** specimens studied here (Fig. 1). Therefore, I consider those paratypes with smooth tentacles to be juveniles of *A. paulayi* **n. sp.** with underdeveloped buccal tentacles and abdominal neuropodial cirri.

Specimens of *A. paulayi* **n. sp.** were previously identified as *A. acutifrons* (*Grube, 1860*) or *Ampharete*. However, the presence of 12 abdominal segment, abdominal pinnules with elongated neuropodial cirri and the spiral arrangement of paleae of *A. paulayi* **n. sp.** differs from *A. acutifrons*, which has 11 abdominal segments, entire abdominal pinnules without neuropodial cirri, and a semicircular arrangement of paleae (*Krüger et al., 2022*).

*Ampharete paulayi* **n. sp.** (type locality: Washington) resembles *A. grubei* Malmgren, 1865 (TL: Svalbard, Arctic Ocean), *A. cirrata* *Webster & Benedict, 1887* (TL: Maine, NE Atlantic), and *A. brevibranchiata* *Treadwell, 1926* in the presence of 12 abdominal segments and pygidial cirri surrounding the anus. *Ampharete brevibranchiata*, described from the Bering Strait, differs in having a rounded neuropodial cirri, rather than elongate cirri as in the other three species. The presence of elongate cirri on the last thoracic neuropodia separates *A. cirrata* from *A. paulayi* **n. sp.** and *A. grubei*. *Ampharete paulayi* **n. sp.** additionally differs from *A. grubei* in the body size of specimens, the width of the interbranchial gap, and the size of pygidial cirri. Complete specimens of *A. paulayi* **n. sp.** (15–33 mm long, 1.7–5 mm wide) are smaller than those of *A. grubei* (27–50 mm long, 4.5–9 mm wide), have a shorter interbranchial gap (almost as wide as the branchial width, instead of 1.5–2 times wider than the branchiae as in *A. grubei*), and have longer and slender pygidial cirri (*Krüger et al., 2022*, fig. 11D).

### *Ampharete cirrata* *Webster & Benedict, 1887*
Figures 6–10
*Ampharete cirrata* *Webster & Benedict, 1887*: 747, pl. 8, figs. 110–112. Type locality: Gulf of Maine.

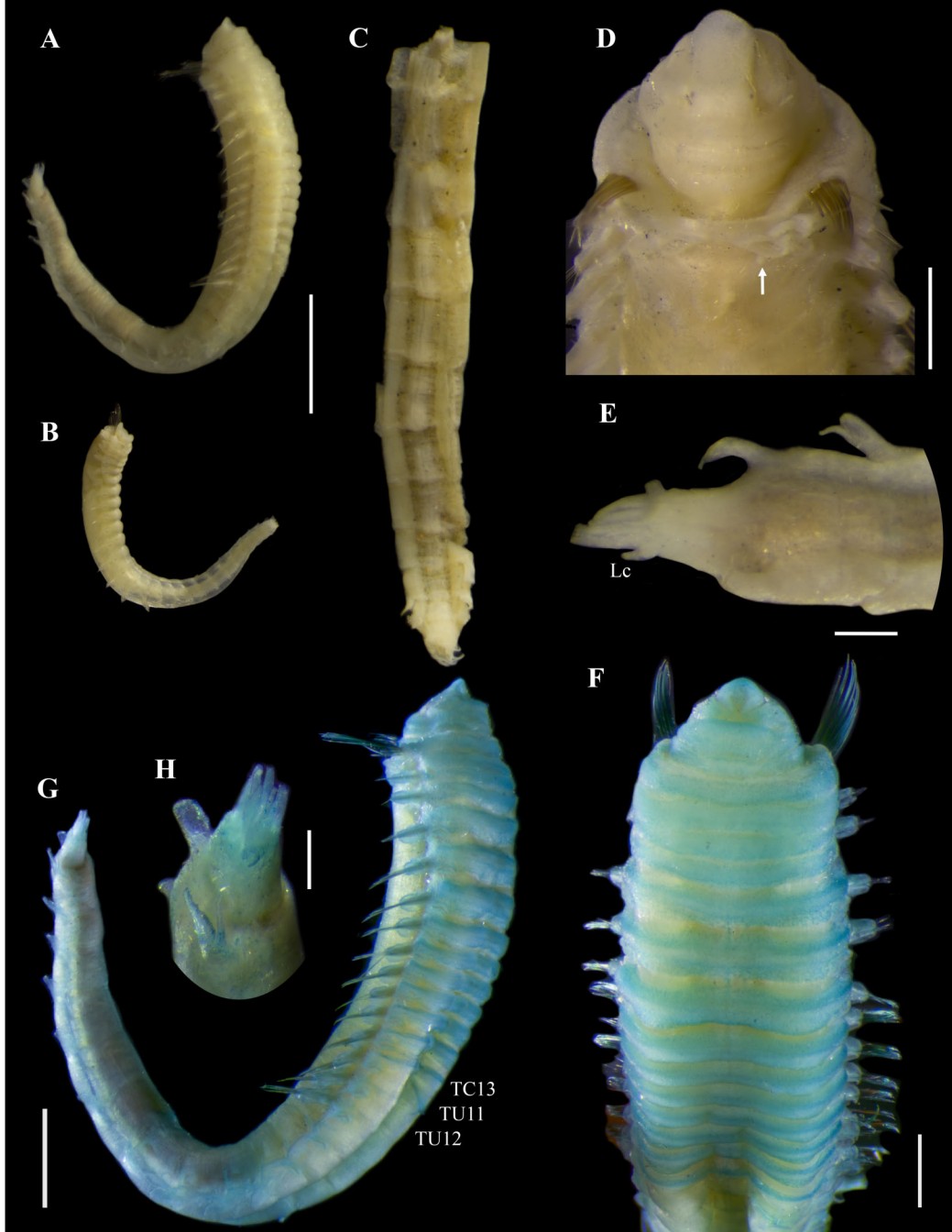

**Figure 6** *Ampharete cirrata Webster & Benedict, 1887*, **syntypes (USNM 457).** (A) Described syntype, lateral view. (B) Smallest syntype, lateral view. (C) Incomplete syntype, only posterior abdomen. (D) Anterior region, dorsal view (arrow points dorsal nephridial papilla). (E) Pygidium, dorsal view. (F–H) Syntype after Methyl green. (F) Anterior region, ventral view. (G) Complete specimen, lateral view. (H) Pygidium, latero-frontal view. Scale bars: A–C: 2 mm; D, F: 0.5 mm; E, H: 0.2 mm; G: 1 mm.

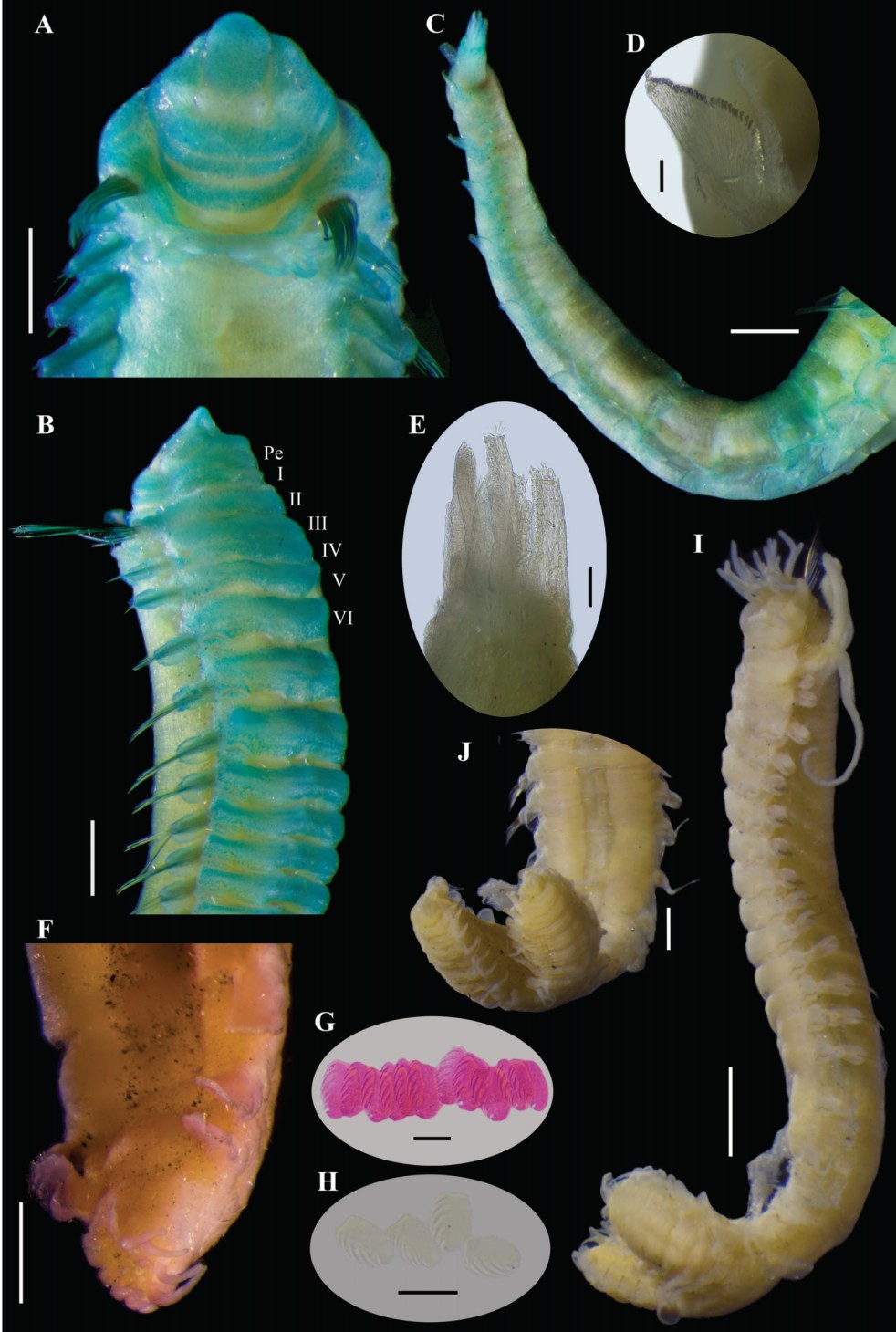

**Figure 7** *Ampharete cirrata* *Webster & Benedict, 1887*, **syntype (USNM 457).** (A) Anterior region after Methyl green staining, dorsal view. (B) Same, lateral view. (C) Abdomen, lateral view. (D) Last abdominal neuropodia. (E) Pygidial cirri, lateral view. (F) Posterior region of incomplete syntype after Shirlastain–A staining, ventral view. (G) Anterior uncini (**Slide 1033**). (H) Posterior uncini (**Slide 1035**). (I) Complete non-type specimen (**USNM 44343**) with two posterior abdominal regions, lateral view. (J) Same, close-up of posterior region, ventral view. Scale bars: A–C, F, J: 0.5 mm; D, G–H: 20 μm; E: 50 μm; I: 1 mm.

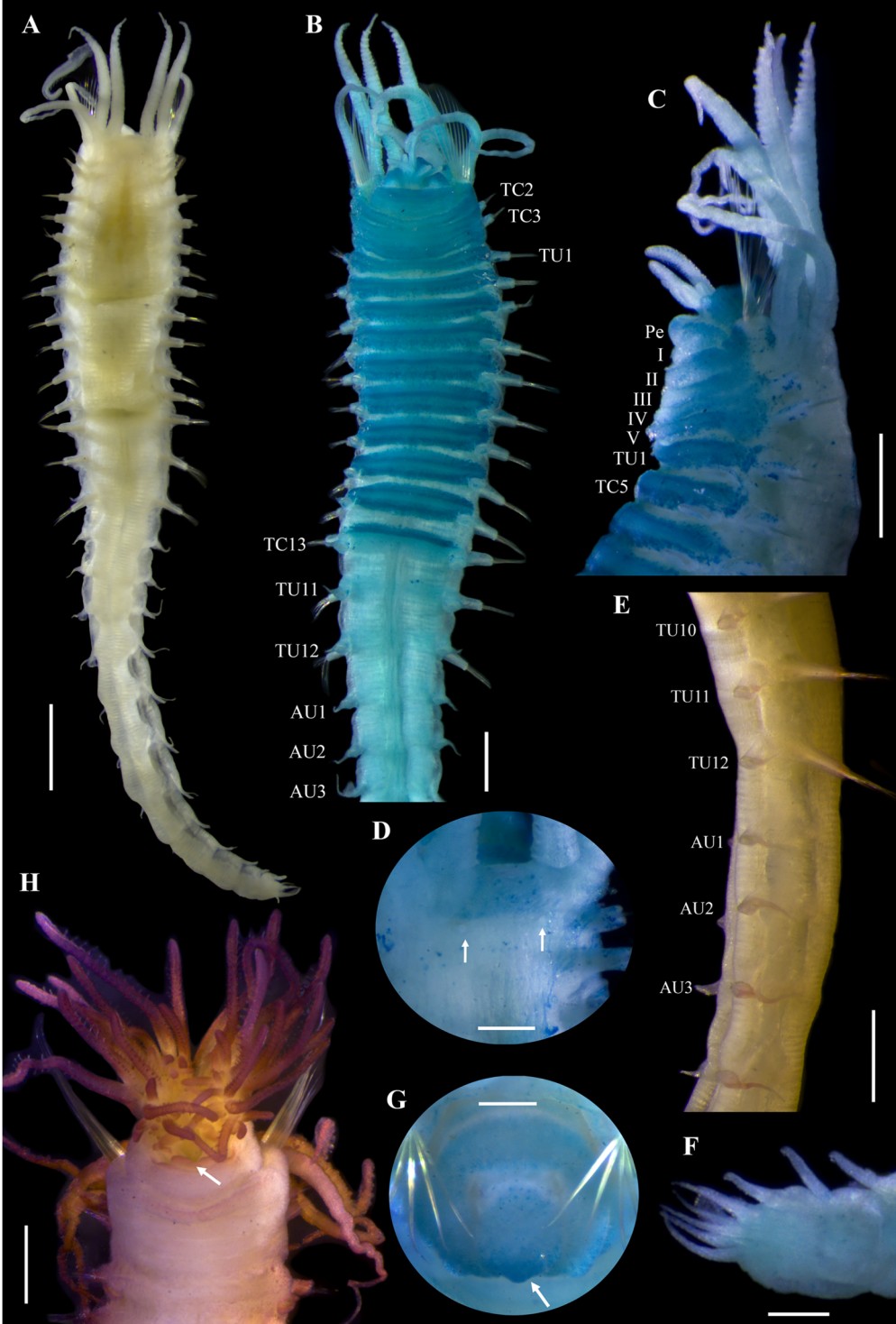

**Figure 8** *Ampharete cirrata* **from Washington (UF 8082).** (A) Complete specimen, dorsal view. (B) Thorax after Methyl green staining, ventral view. (C) Anterior region, lateral view. (D) of post-branchial region, dorsal view (arrows point dorsal nephridial papillae). (E) Transition thorax-abdomen after Shirlastain-A staining, lateral view. (F) Pygidium, lateral view. (G) Prostomium after Methyl green staining, frontal view (**UF 8098**; arrow points intensely stained region). (H) Anterior region, ventral view (**UF 8071**; arrow points to ventral pharyngeal organ). Scale bars: A: 1 mm; B–C, E, H: 0.5 mm; D, F–G: 0.2 mm.

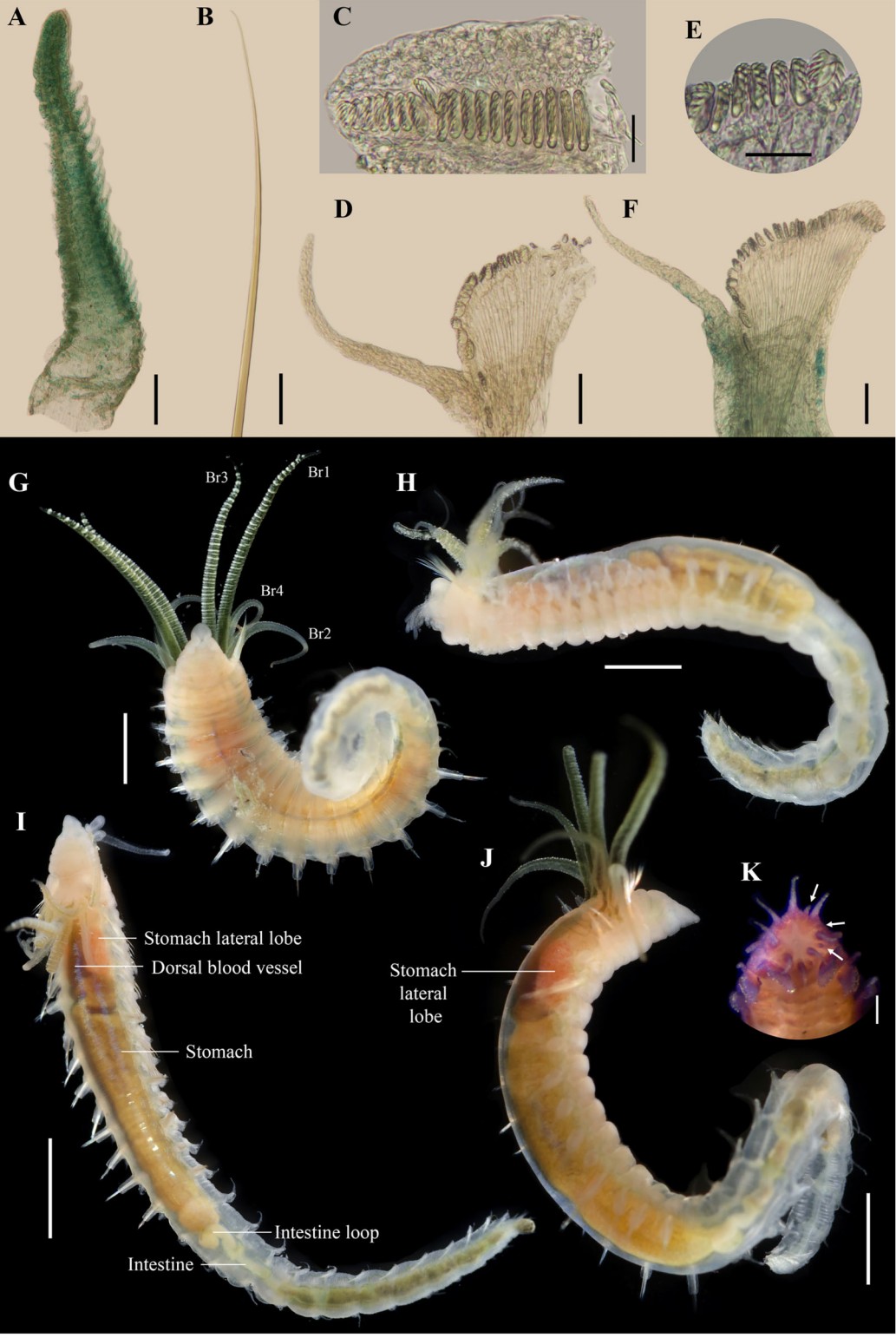

**Figure 9** *Ampharete cirrata* **from Washington (UF 8082).** (A) Buccal tentacle after Methyl green staining. (B) Palea. (C) Thoracic unciniger 4 (seg. IX). (D) Abdominal pinnules from AU4. (E) Close-up of abdominal uncini. (F) Abdominal pinnules AU4 after Methyl green staining (**UF 8081**). (G–J) Specimens in life. (G) **UF 8068** spec., ventral view. (H) **UF 8071** spec., lateral view. (I) **UF 8081** spec., dorsal

**Figure 9** (continued)
view (some internal organs visible by transparency). (J) **UF 8069** spec., dorso-lateral view. (K) Pygidium after Shirlastain-A staining, frontal view (**UF 8069**; arrows point to intercalated inconspicuous cirri). Scale bars: A–B, K: 100 µm; C, E: 20 µm; D, F: 50 µm; G–H, J: 1 mm; I: 2 mm. Photos G–J credit: Gustav Paulay.           

**Diagnosis.** *Ampharete* with 9–14 paleal chaetae per side on segment II. Neuropodial cirri from TU5–6. Anterior thoracic neuropodia with rounded dorsal cirri. Last 4–5 thoracic segments (from TU9–TU10) with elongated neuropodial cirri, protruding beyond tori length. Abdomen with 12 segments. All abdominal neuropodia with elongated dorsal cirri protruding beyond pinnules length. Pygidium with two lateral cirri and 8–18 long cirri surrounding the anus.

**Material examined**

**Syntypes**

MAINE • 3 spec., 4 slides; Eastport, North Atlantic; depth 11–18 m; Webster H. leg.; **USNM 457**.

**Additional material**

MAINE • 60 spec.; Muscongus Bay, near Hog Island, Audubon Camp; depth 5.4 m; mud; 29 Aug. 1955; M.H. Pettibone leg.; **USNM 44343**.

WASHINGTON • 1 incomplete spec.; San Juan County, Reads Bay; 48.49626°N, 122.82139°W; depth 8.5 m; soft bottom; 24 Apr. 2019; Gustav Paulay leg.; BOLD COI: BBPS506-19 (identified as *A. acutifrons*); **UF 8068** • 1 incomplete spec.; same data as for preceding; BOLD COI: BBPS507-19 (identified as *A. acutifrons*); **UF 8069** • 1 incomplete spec.; same data as for preceding; BOLD COI: BBPS508-19 (identified as *A. acutifrons*); **UF 8071** • 1 incomplete spec.; same data as for preceding; BOLD COI: BBPS509-19 (identified as *A. acutifrons*); **UF 8081** • 2 complete spec.; same data as for preceding; **UF 8082** • 3 complete spec.; same data as for preceding; **UF 8098**.

BALTIC SEA • 1 incomplete spec.; S Baltic Sea; 55°N, 14.6°E; depth 46 m; Örbeg leg.; **SMNH-120823**.

DENMARK • 3 spec.; South of Helsingor, Oresund; depth 18 m; 24 May. 1961; Eliason leg.; **USNM 43292**.

NORWAY • 2 spec.; Leirbugt, Posangerfjord; depth 10 m; 11 July 1939; G.I. Crausford leg.; **BMNH 1950.2.15.30**.

WESTERN RUSSIA • 8 spec.; Novaya Zemlya, Matotschkin scharr, western opening, Arctic Ocean; Jenissej exped. 1875, sta. 80; 72°30′N, 54°40′E; depth 4–9 m; mud; **SMNH-120734**.

**Redescription.** Syntype complete (**USNM 457**), body pale yellow, 9.5 mm long, 1.5 mm wide (Fig. 6A). Prostomium subdivided into two lobes, middle lobe anteriorly rounded with a pair of eyespots below the ciliated pits of the nuchal organs (Figs. 6D, 7A). Buccal tentacle inside the mouth, not visible. Lower lip entire (Fig. 6F).

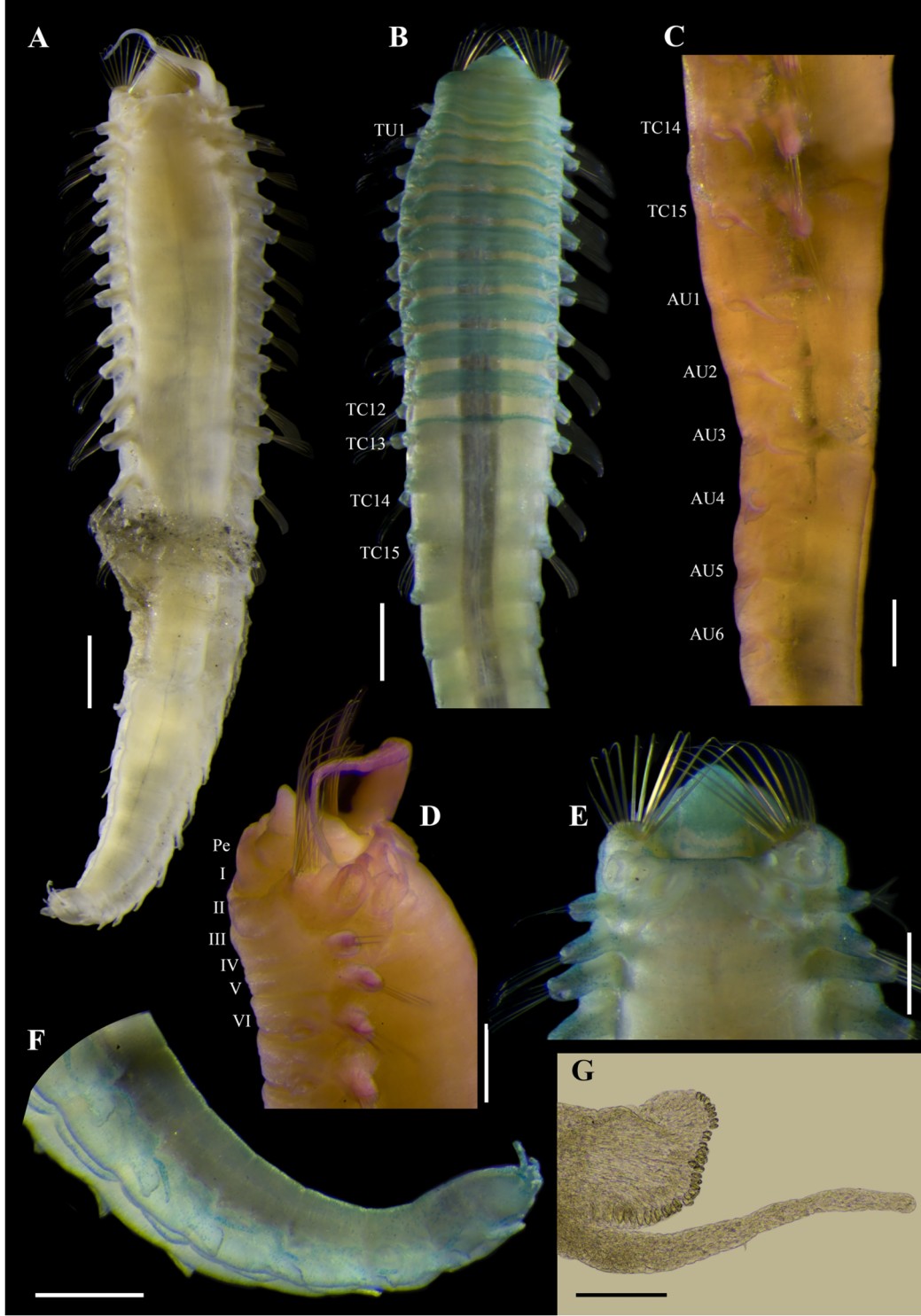

**Figure 10** *Ampharete cirrata* **from Novaya Zemlya (SMNH 120734).** (A) Complete specimen, dorsal view. (B, E, F) Specimen after Methyl green staining. (B) Thorax, ventral view. (C, D) Specimen after Shirlastain-A staining. (C) Transition thorax-abdomen, lateral view. (D) Anterior region, lateral view. (E) Same, dorsal view. (F) Posterior abdomen and pygidium. (G) Abdominal unciniger 6. Scale bars: A–B: 1 mm; C–F: 0.5 mm; G: 100 μm.

Thorax with 17 segments: 15 chaetigers, 12 uncinigers (Fig. 6G). Segment I achaetous, clearly distinguishable in ventral view, not fused with peristomium (Fig. 7B). Segment II with paleae arranged in a semicircle: 12 chaetae on the right side, 11 chaetae on the left. Paleal chaetae bright golden, slender, gradually tapering with caudate tips. Paleal chaetae tips very fragile, most paleae with broken tips. Paleae 0.9 mm long, as long as prostomium, about three times longer than first thoracic notopodium (notochaetae broken).

Four pairs of branchiae originating from segments II–V. Branchiae arranged in two transverse rows. Three pairs of branchiae in first row on segments II–III, with a median gap shorter than branchial scars width, 4th pair of branchiae shifted behind middle branchiae (Br1) of anterior row with bases extended to segment V (TC3). All branchiae lost, only scars visible. Two nephridial papillae between 4th pair of branchiae (Fig. 6D). Nephridial papillae well separated by a gap of 0.3 mm, ~8 times as long as width of nephridial papillae.

Notopodia with capillary chaetae on 14 segments from segment IV. Segment III achaetous, fused dorsally with segment II, easily distinguishable in ventral and lateral view, no rudimentary notopodium visible. Notopodia lobes of similar size, twice longer than wide. Notochaetae arranged in two rows, each with 5–6 bilimbate chaetae. Several notochaetae broken in the first four notopodia. Nephridial papillae below notopodial lobes not observed.

Neuropodia tori present on 12 segments from chaetiger 4 (segment VI; Fig. 7B). Thoracic neuropodia with neuropodial cirri from TU8. Neuropodial cirri short and rounded on TU8–10. Posterior thoracic neuropodia (TU11–12) with an abruptly elongated cirri, ~9 times longer than wide, as long as neuropodia width. Ventral shields along chaetigers 1–13 (segment XV; Figs. 6F, 6G). Continuous median ventral groove visible from chaetiger 14 (segment XVI, TU11) to posterior end.

Abdomen with 12 segments (Fig. 7C). First two segments with intermediate uncinigers. Pinnules from the third abdominal segment. All abdominal neuropodia with elongated dorsal cirri. Neuropodial cirri progressively longer along posterior segments until AU6–AU7, which have the longest cirri (~11 times longer than wide), then progressively shorter until be ~6 times longer than wide (Fig. 7D).

Pygidium with two lateral cirri and about eight long cirri surrounding anus in one row (Figs. 6E, 6H). Pygidial cirri elongate, ~0.2 mm long, 7–9 times longer than wide (Fig. 7E).

**Methyl green staining pattern.** Prostomium completely stained, except for margins of middle lobe and a transverse band including eyespots and nuchal organs (Fig. 7A). Peristomium and segment I completely stained, except for junctional regions dividing each ring (Figs. 7A, 7B). All neuropodial tori slightly stained, surrounding uncini row. Ventral shields strongly stained along TC1–13 (TU10; Figs. 6F, 6G, 7A). Neuropodial cirri not stained (Figs. 6G, 7C). Lateral cirri on pygidium slightly stained, surrounding cirri non-stained (Fig. 6H).

**Variation syntype specimens.** The type series of *A. cirrata* consists of three syntypes (**USNM 457**): two complete and one incomplete specimen (Figs. 6A–6C). The longest, complete syntype was used to redescribe the species (Figs. 6A, 6D–6F). Smallest, complete

syntype 5 mm long, 0.7 mm wide, with 12 abdominal segments (Fig. 6B); neuropodial cirri distinguishable along TU7–AU12, cirri small and rounded on TU7–9, cirri slightly elongated along TU10–AU1, and long cirri protruding the pinnules length from AU2 to the last abdominal segment; pygidium and pygidial cirri not distinguishable. Incomplete syntype consisting of only posterior region of 10 mm long, 1.5 mm wide, with 11 abdominal segments (Fig. 6C); pygidium with a pair of long lateral cirri and ~12 long surrounding cirri (4–5 times longer than wide), about three surrounding cirri lost (Fig. 7F).

There are four slides labeled as types of *A. cirrata*. **Slide 1032** includes anterior capillary chaetae. **Slide 1034** has thoracic parapodia. **Slides 1033** and **1035** have anterior and posterior uncini, respectively; therefore, no additional dissection of uncini was made of the complete syntype used in the redescription. Anterior and posterior uncini only visible in lateral view, with 5–6 denticles (Figs. 7G, 7H).

**Variation non-type specimens.** Sixty specimens from Maine (**USNM 44343**). Complete specimens of 8.5–19 mm long, 1.1–1.9 mm wide, with bipinnate tentacles, longest paleae 0.9–1.6 mm long, 12 abdominal segments, and about 10–12 long pygidial cirri. One specimen with posterior region bifurcated, probably the result of a mutation: each posterior abdominal regions with 8–9 AU and pygidium with lateral and surrounding cirri (Figs. 7I, 7J).

Nine specimens from Washington: five complete specimens and four incomplete with body split into two regions (probably due to tissue sampling for DNA extraction). Complete specimens 9–11 mm long, 1.1–1.4 mm wide (Figs. 8, 9); with bipinnate tentacles (Fig. 9A), 12–14 paleal chaetae per side (Figs. 8A–8C, 9B); thoracic neuropodia with elongate dorsal cirri from TU8–TU10 (Figs. 8A, 8B, 8E); 12 abdominal segments, all of them with long neuropodial cirri (Figs. 8E, 9D). All specimens with lateral and surrounding pygidial cirri, 8–9 surrounding long cirri interspersed with 8–9 short cirri (Figs. 8F, 9K). Neuropodial and pygidial cirri not stained by Methyl green (Figs. 8F, 9F).

Fifteen specimens from northeastern Atlantic Ocean with neuropodial and pygidial cirri stained with Methyl green. One incomplete specimen from the Baltic Sea, body 13 mm long, 2.1 mm wide with complete thorax and only 5 AU (**SMNH-120823**). Three specimens from Denmark (**USNM 43292**): two complete juvenile (5–7 mm long, 0.6–0.8 mm wide, with 12 AU, smooth tentacles, and 9–11 paleal chaetae per side, longest paleae of 0.5–0.7 mm long) and one incomplete specimen (9.5 mm long, 1.6 mm wide, with complete thorax and only 8 AU, bipinnate tentacles, 13–15 paleal chaetae per side, longest paleae of 1.2 mm long). Two complete, mature female specimens from Norway of 11–17 mm long, 1.3–2 mm wide with 12 AU, longest paleae 0.8–1.5 mm long (**BMNH 1950.2.15.30**). Eight specimens from Novaya Zemlya (**SMNH-120734**): seven complete specimens (9–12.5 mm long, 1.6–1.7 mm wide with 12AU, ~12 paleal chaetae per side, longest paleae of 1.2 mm long) and one incomplete (9 mm long, 1.8 mm wide with complete thorax and only 3 AU).

**Body coloration in life.** Specimens from Washington (**UF 8068**, **UF 8069**, **UF 8071**, **UF 8081**) with translucent, light melon body color. Branchiae green or orange with white horizontal lines (Figs. 9G, 9H). Paleae yellowish. Some internal organs visible through

transparent body wall: dorsal blood vessel dark reddish, anterior lobe of stomach pinkish (Figs. 9I, 9J), stomach pale yellowish, intestine colorless (Fig. 5I). Abdomen colorless, translucent (Figs. 9G–9J).

**Remarks.** *Ampharete cirrata Webster & Benedict, 1887* was originally described based on specimens from Maine, USA collected from sediments at 11–18 m depth. Although the original description of *A. cirrata* is brief (*Webster & Benedict, 1887*: 747), it includes the relative size of branchiae (referred to as long as body width or slightly longer), the presence of neuropodial cirri from 9th chaetiger segment (TU6), 14 pygidial cirri, body coloration (body light green and branchiae light green with white transverse bands on ventral surface), and the size of the body (24 mm long, 4 mm wide) and branchiae (4 mm long) of the largest specimen. The two complete syntypes of *A. cirrata* (**USNM 457**) match Webster & Benedict's description. The illustrations of *A. cirrata* by *Webster & Benedict (1887*: pl. 8, figs. 110–112) appears to be based on the largest and incomplete specimen, which is the only one with neuropodia dissected. Therefore, it is likely that the four slides were made from this incomplete syntype, which lacks most abdominal neuropodia. However, since it is an incomplete specimen with only 11 abdominal segments and pygidium, the best-preserved specimen of *A. cirrata* is redescribed here. *Krüger et al. (2022)* provided two micrographs of the same syntype specimen redescribed here.

The syntypes of *A. cirrata* and non-type specimens examined from the NE Pacific, NE Atlantic and the Arctic Ocean have notable morphological similarities, including 12 abdominal segments, neuropodial cirri on thoracic and abdominal segments, and long pygidial cirri surrounding the anus. However, specimens from Washington (*e.g.*, **UF 8068**, **UF 8069**) differ from *A. cirrata sensu stricto* in the number of pygidial cirri and in the body coloration. Syntypes of *A. cirrata* have a pygidium with 8–12 long surrounding cirri of similar size (Figs. 3E, 3F) and a light green live body coloration, rather than 16 surrounding pygidial cirri (eight long cirri and eight short cirri interspersed; Fig. 5K), and a light melon-colored, translucent body (Figs. 5G–5J) as in the Washington specimens. However, there are no differences in other characteristics, including the branchiae coloration pattern of the living specimens.

Specimens from Russia (**SMNH-120734**; Fig. 10), Norway (**BMNH 1950.2.15.30**), Denmark (**USNM 43292**) and the Baltic Sea (**SMNH-120823**) differ from *A. cirrata* syntypes and Washington specimens in the intense staining of the neuropodial and pygidial cirri with Methyl green (Fig. 10F). The Methyl green staining pattern of the northwest Atlantic specimens matches the records of *Krüger et al. (2022)* of *A. cirrata* in the Baltic Sea with specimens collected at 44–48 m depth.

There are no COI sequences of *A. cirrata* from Maine or the northwestern Atlantic, but molecular analysis revealed K2P-corrected distances of 4.4–4.9% between the Washington and the Baltic Sea populations, in contrast to the interspecific distance of 14.4–22.9% observed among morphological distinct *Ampharete* specimens (Tables S2, S3). The subtle morphological differences observed among the specimens of *A. cirrata* examined here, and the relatively low genetic divergence between Washington and Baltic Sea specimens suggest recent population isolation. However, separating the NE Pacific and NE Atlantic populations into different species could lead to confusion and identification difficulties,

given the strong morphological resemblance to *A. cirrata*. Obtaining COI sequences for specimens of *A. cirrata* from Maine or the northeast Atlantic and reviewing more material from the Arctic Ocean and surrounding areas could clarify the distribution range of *A. cirrata* sensu stricto.

### *Ampharete labrops* Hartman, 1961
Figures 11–13
*Ampharete labrops* Hartman, 1961: 127–128, pl. 34, figs. 1–4. Type locality: Santa Barbara Point, California, 9.14 m, from a bottom of fine sand with kelp.

**Diagnosis**. *Ampharete* with anterior margin of prostomium with rows of tiny eyespots. Segment II with paleae, 9–11 paleal chaetae per side. All thoracic neuropodia entire, without cirri. Abdomen with 13 segments. Abdominal pinnules (AU3–13) with neuropodial cirri, as long as pinnules length; cirri progressively shorter towards posterior segments. Pygidium with one pair of long lateral cirri, surrounding cirri absent.

**Material examined**
**Holotype**
CALIFORNIA • 1 incomplete spec.; 2 miles from Santa Barbara Point light; 34.4041°N, 119.7611°W; depth 9.14 m; R/V Velero IV, Sta. AHF 6694–59; 3 Dec. 1959; Allan Hancock Foundation leg.; dark gray to black fine sand with worm tubes, kelp, and much woody debris; **LACM-AHF Poly 267**.

**Additional material**
WASHINGTON • 1 incomplete spec.; Jefferson County, Port Townsend, mouth of Kilisut Harbor; 47.09354°N, 122.73316°W; 1 Apr. 2019; depth 4 m; Gustav Paulay leg.; soft bottom; BoldSystems COI gene: BBPS810-19; **UF 8202**.
CALIFORNIA • 2 spec.; Orange County, off Oceanside; 33.2130°N, 117.4188°W; 18 Jul 1994; depth 13 m; soft sediment; Van Veen grab; **LACM-AHF Poly 12845** • 33 spec.; Los Angeles County, Palos Verdes Peninsula, off San Pedro; 33.704444°N, 118.310278°W; depth 21.95 m; R/V Velero IV, Sta. AHF 5028-57; 24 Apr. 1957; Allan Hancock Foundation leg.; coarse gray and black sand; Hayward Orange Peel grab; **LACM-AHF Poly 13546**.

**Description.** Holotype incomplete (**LACM-AHF Poly 267**), mature female, body stained blue, 11 mm long, 2 mm wide (Figs. 11A, 11B). Prostomium subdivided into two lobes, middle lobe anteriorly straight, with one pair of eyespots below nuchal organs (as ciliated pits; Fig. 11C). Anterior margin of prostomium with two or more rows of tiny brown eyespots, except in central slightly pointed region (Fig. 11D). Buccal tentacles inside mouth, partially visible, bipinnate. Lower lip entire (Fig. 11D).

   Thorax with 17 segments: 15 chaetigers, 12 uncinigers. Segment I clearly distinguishable in ventral and lateral view, not fused with peristomium (Fig. 11D). Segment II with paleae arranged in semicircle: 12 paleal chaetae per side (Fig. 11C). Paleal chaetae bright golden, thick, flat, gradually tapering with caudate tips (Fig. 12A). Paleae 0.3–0.7 mm long, longest

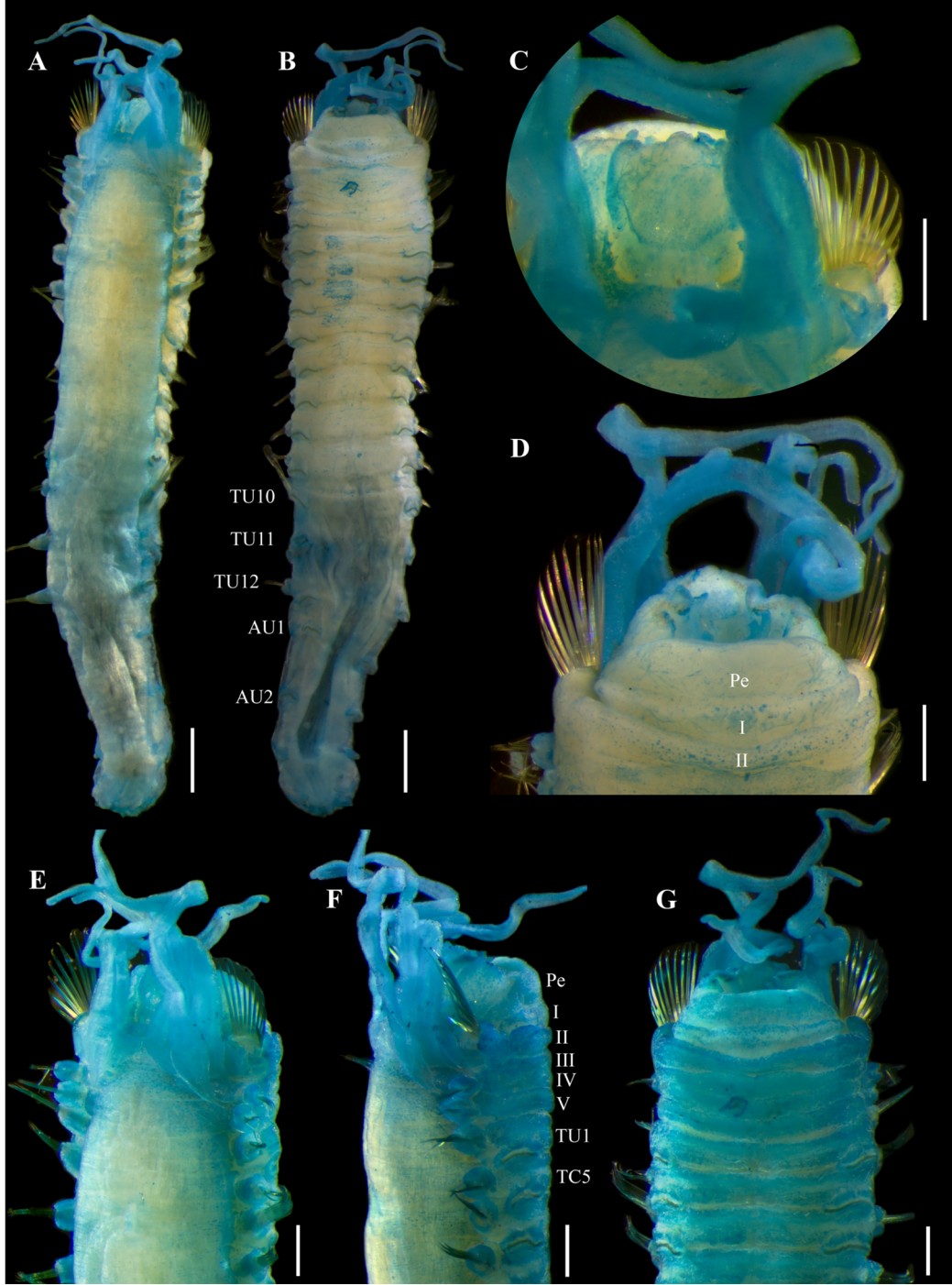

**Figure 11** *Ampharete labrops Hartman, 1961*, holotype ( LACM-AHF Poly 267 ). (A) Incomplete specimen, dorsal view. (B) Same, ventral view. (C) Middle lobe of prostomium, dorsal view. (D) Anterior region, ventral view. (E) Anterior region after Methyl green staining, dorsal view. (F) Same, lateral view. (G) Same, ventral view. Scale bars: A–B: 1 mm; C–G: 0.5 mm.

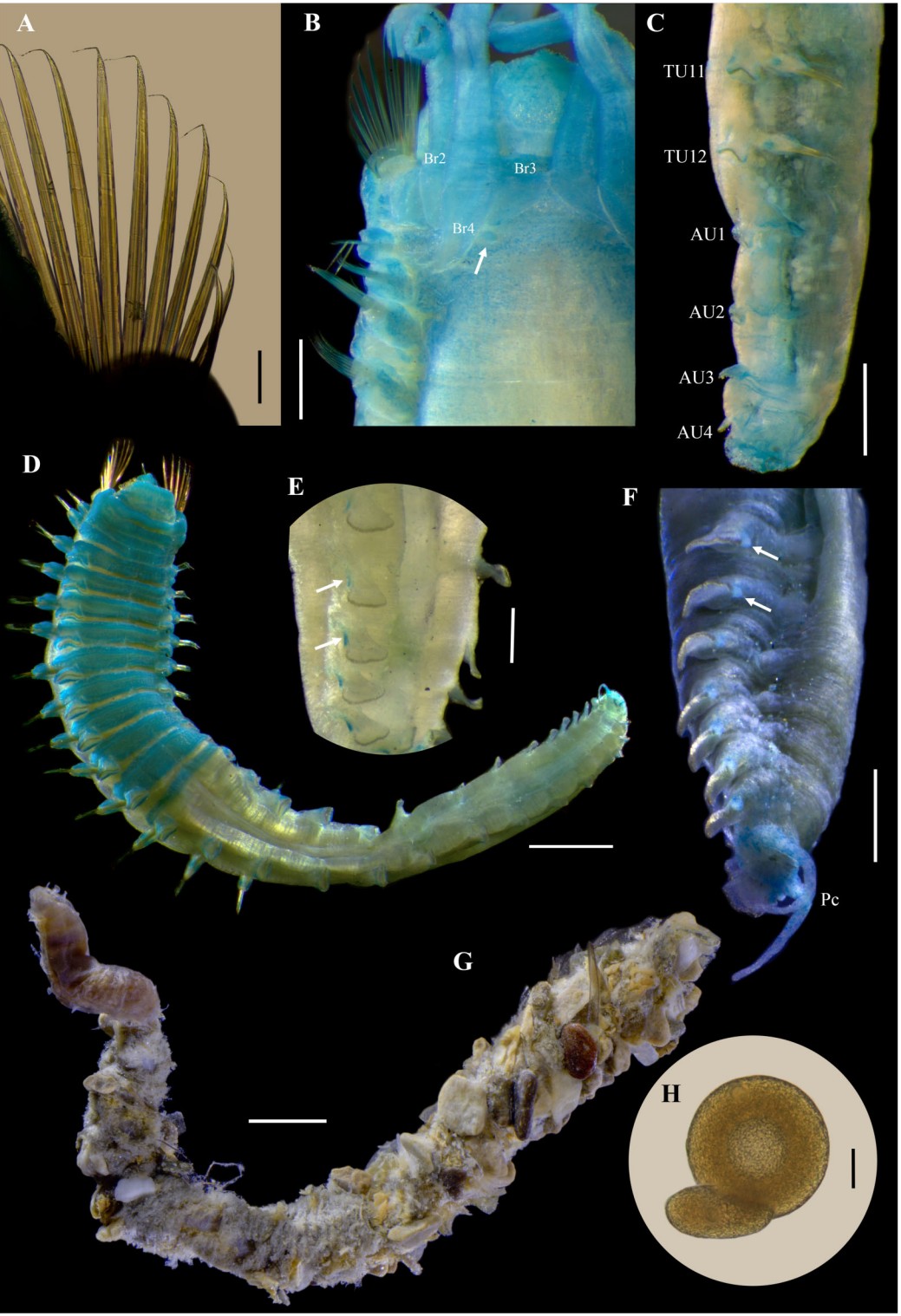

**Figure 12** *Ampharete labrops Hartman, 1961*, **holotype (LACM-AHF Poly 267).** (A) Paleal chaetae.
(B) Branchial arrangement (arrow points nephridial papilla), dorsal view . (C) Posterior region, lateral
view. (D) Complete non-type specimen (**LACM-AHF Poly 126952**), ventral view. (**E**) Same, close-up of

**Figure 12 (continued)**
abdominal neuropodia (arrows point dorsal lobe). (F–H) Non-type specimens (**LACM-AHF Poly 13546**). (F) Posterior region, pygidium with long lateral cirri (Pc). (G) Complete specimen inside the tube. (H) Oocytes. Scale bars: A: 100 μm; B, F: 0.5 mm; C: 1 mm; D, G: 2 mm; E: 0.3 mm; G: 20 μm.

paleae protruding beyond prostomium, twice longer than TC2 notochaetae (0.3 mm), and almost as long as longest notochaetae (TC6, 0.6 mm).

Four pairs of branchiae originating from segments II–V (Figs. 11E, 11F). Branchiae arranged in two transverse rows. Three pairs of branchiae arranged in a transverse row on segment II–III, without median gap (Fig. 12B), 4th pair shifted behind innermost (Br3) and middle (Br1) branchiae of transverse row with bases extended to segment V (TC3). Branchiae filiform, 1.7–4.5 mm long (Figs. 11E–11G); outermost branchia (Br2) longest, reaching back to thoracic chaetiger 11 (TU8). Three branchiae lost, only scar remains: Br1 and Br3 left side, Br2 right side. Two dorsal nephridial papillae between 4th pair of branchiae (Fig. 12B). Nephridial papillae well separated by a wide gap of 0.7 mm, ~6 times as long as nephridial papillae width (Fig. 12B).

Notopodia with capillary chaetae on 14 segments from segment IV (Fig. 11A). Segment III fused dorsally with segment II, easily distinguishable on ventral and lateral view, with an inconspicuous rudimentary notopodia without chaetae. Notopodia 2–3 times longer than wide, lobe progressively longer towards posterior region. Notochaetae arranged in two rows, each with 7–8 bilimbate chaetae. Nephridial papillae below notopodial lobes not observed.

Neuropodia tori present on 12 segments from chaetiger 4 (segment VI) (Figs. 11F, 11G). Thoracic neuropodia smooth, without cirri. Ventral shields along chaetiger 1–13 (segments II–XV) (Fig. 11B). Continuous median ventral groove visible from chaetiger 14.

Abdomen incomplete, only 4 chaetigers (Fig. 12C). Small and rounded tubercle (rudimentary notopodia?) along abdominal segments 1–4. First two segments with intermediate uncinigers. Third and four abdominal segments with pinnules. Abdominal pinnules with slightly elongated neuropodial cirri, smaller than pinnules length.

**Methyl green staining pattern of holotype.** Prostomium, peristomium and segment I completely stained (Figs. 11E–11G). Anterior margin of prostomium intensely stained. Few spots in dorsal region of thorax, near nephridial papillae (Figs. 11E, 12B). Thoracic notopodia and neuropodia with surrounding spots (Figs. 11F, 11G). Notopodia with a small, rounded lateral spot stained (Fig. 12C). Ventral shields strongly stained from TC1–13.

**Variation.** Complete specimens (**LACM-AHF Poly 126952, LACM-AHF Poly 13546**) 7–25 mm long, 1.2–2.7 mm wide, body white to pale brown, 8–13 paleal chaetae per side (0.5–1.1 mm long), 13 abdominal segments (rarely 14 AU), two long lateral anal cirri, reaching forward to the last 2–4 abdominal segments. Non-type specimens stained with Methyl green (Figs. 12D–12F): ventral shields strongly stained from TC1–13 (Fig. 12D), abdominal pinnules with the base of neuropodial cirri stained, tip non-stained (Fig. 12E),

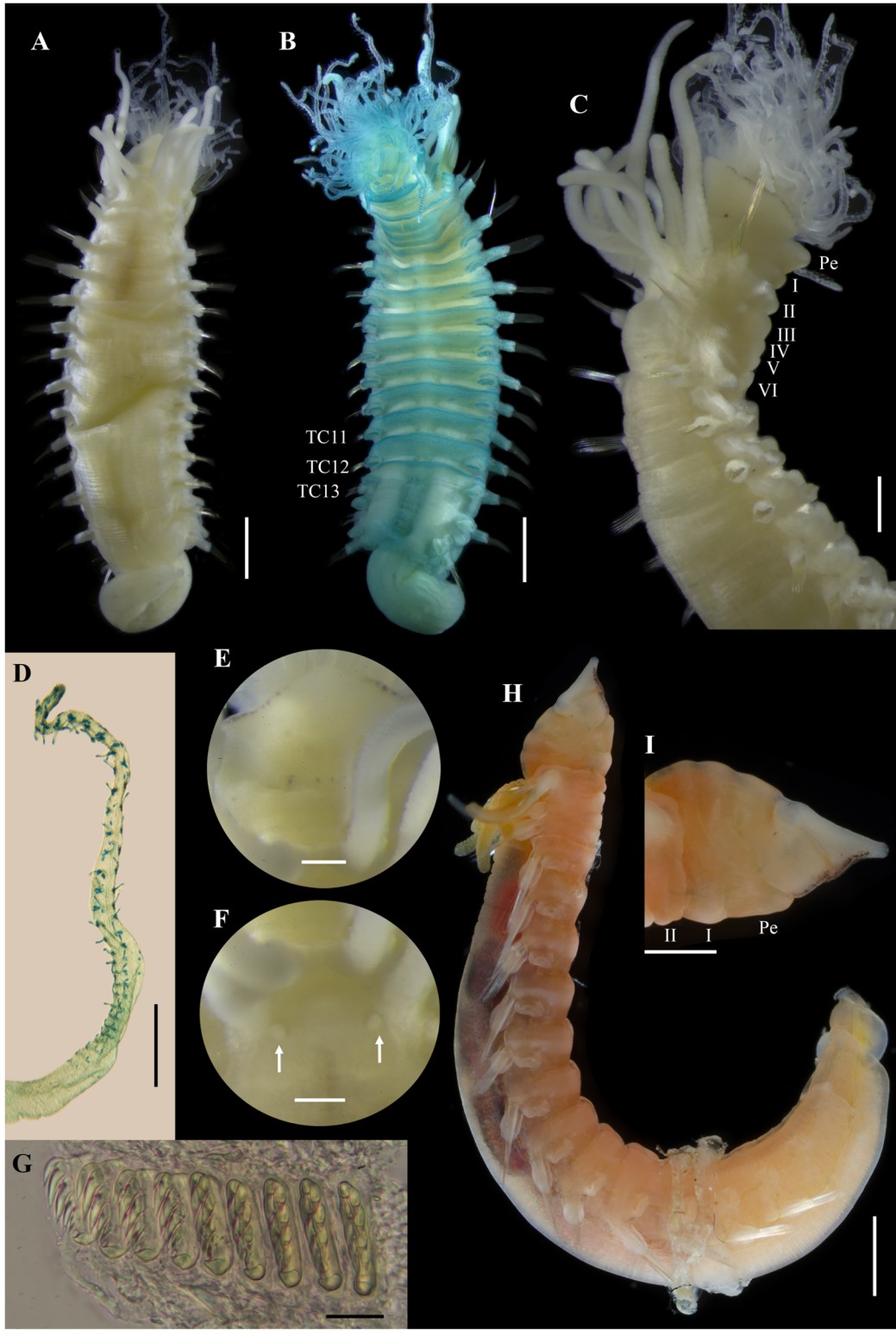

**Figure 13** *Ampharete labrops* **from Washington (UF 8202).** (A) Incomplete specimen, dorsal view. (B) Same after Methyl green staining, ventral view. (C) Anterior region, lateral view. (D) Tentacle stained with Methyl green. (E) Close-up of prostomial eyespots. (F) Close-up of post-branchial region

**Figure 13 (continued)**
(arrows point nephridial papillae). (G) Thoracic uncini (TU4), frontal view. (H) Specimen in life. (I) Same, close-up of head; anterior margin of prostomium with tiny eyespots. Scale bars: A–B, H: 1 mm; C, I: 0.5 mm; D: 300 μm; E–F: 200 μm; G: 20 μm. Photos H–I credit: Gustav Paulay.

pygidial lateral cirri strongly stained (Fig. 12F). Some specimens inside their tubes (Fig. 12G). Mature females with oocytes of ~70 μm diameter (Fig. 12H).

Incomplete specimen from Washington (**UF 8202**; Fig. 13) with prostomial middle lobe anteriorly pointed (Figs. 13A, 13E, 13H, 13I), with two pairs of eyespots (Fig. 13E). Buccal tentacles everted (Fig. 13A–13C), bipinnate, with only two rows of short filaments (Fig. 13D); tentacles barely stained with methyl green, except for filaments and their bases which are strongly stained. Dorsal nephridial papillae separated by a gap of 0.3 mm, 5 times the width of the nephridial papillae (Fig. 13F). Ventral side of branchial surface with transverse rows of cilia, except in the outermost pair (Br2) of branchiae. Thoracic uncini with two rows of denticles in front view, 4–5 denticles over rostral tooth in lateral view, with subrostral process (Fig. 13G).

**Body coloration in life.** Incomplete specimen (**UF 8202**) with body pinkish orange, abdomen with a lighter shade (Fig. 13H). Prostomium translucent with dark marginal eyespots, except in center region of middle lobe (Fig. 13I). Peristomium and first two segments pinkish orange. Ventral shields slightly white. Branchiae orange. Stomach dark reddish, visible through transparency. Thoracic notochaetae colorless.

**Remarks.** *Ampharete labrops Hartman, 1961* was originally described based on an incomplete specimen collected from California at 9.14 m depth. The original description includes 15 thoracic chaetigers, 12 thoracic uncinigers, four pairs of branchiae, pinnate tentacles, a subquadrate prostomium, 11 paleal chaetae per side on segment II, and the presence of rows of eyespots along anterior margin of the prostomium (*Hartman, 1961*: 127–128). Records of *A. labrops* from California by *Hilbig (2000)* expanded the species diagnosis by noting the presence of 13 abdominal segments and a pygidium with two long lateral cirri, which agrees with the additional specimens of *A. labrops* (**LACM-AHF Poly 12845, LACM-AHF Poly 13546**) from California examined in this study.

Holotype of *A. labrops* is permanently stained blue, probably due to the previous use of Methyl blue, which, unlike Methyl green, is not temporary. The notopodium and neuropodium of chaetiger 13 (segment XV), right side in dorsal view, were previously dissected, but the corresponding slides were not found. No dissections were made from the holotype, to preserve the condition of the specimen.

The presence of marginal eyespots on the prostomium has been the main diagnostic character to distinguish *A. labrops* from other *Ampharete* species (*e.g.*, *Hilbig, 2000*). However, loss of marginal eyespot pigmentation was observed in specimens from lot **LACM-AHF Poly 12845**. I examined these specimens in 2022, and photographs of the prostomial eyespots were taken; however, on a more recent examination (July 2025), there was no trace of the presence of marginal eyespots on either specimen. Some of the possible causes of loss of pigmentation of marginal eyespots could be exposure to sunlight or the

preserving solution (*i.e.*, ethanol 70%). Marginal eyespots can be a useful attribute to quickly identify specimens of *A. labrops*, however, since eyespots pigmentation can be lost, the comparison of other structures is necessary.

The presence of 13 abdominal uncinigers in *A. labrops* is shared with *A. arctica* *Malmgren, 1866*, *A. eupalea Chamberlin, 1920*, *A. finmarchica* (*Sars, 1865*), *A. kudenovi* *Jirkov, 1994*, and *A. seribranchiata Treadwell, 1926*. However, *A. labrops* differs from *A. eupalea* and *A. seribranchiata* by the branchial arrangement into two transverse rows rather than in a single row. The mucronate tips on the paleae of *A. arctica* and *A. finmarchica* separate them from *A. labrops* with filiform paleal tips. The absence of abdominal neuropodial cirri in *A. kudenovi* differs from *A. labrops*, with small and rounded neuropodial cirri.

Although the Washington specimen (**UF 8082**) is incomplete, it exhibits all the diagnostic features of the holotype of *A. labrops*, except for the prostomial shape and the number of eyespots. In the holotype, the anterior margin of the prostomium is straight and slightly pointed in the center (Fig. 11C), whereas in the Washington specimen it is distinctly pointed (Figs. 13E, 13H, 13I). Notably, even when the tentacles are everted, the prostomium of the **UF 8082** specimen retains its pointed shape (Fig. 11E). The Washington specimen has two pairs of eyespots (Figs. 11C, 11E) instead of the single pair described for *A. labrops*. Because these morphological differences are subtle and may be size-related, the specimen is here identified as *A. labrops*.

COI sequences of *A. labrops* from California are not currently available for comparison. However, molecular analysis showed that the incomplete specimen of *A. labrops* examined here (previously identified as *A. acutifrons*) is most closely related to Vancouver Island specimens also identified as *A. labrops* (Fig. 1).

## DISCUSSION

The detailed morphological analysis performed in this study enabled the identification of three *Ampharete* species from Washington: *A. cirrata Webster & Benedict, 1887*, *A. labrops* *Hartman, 1961*, and *A. paulayi* **n. sp**. COI sequences from most Washington specimens examined here were previously deposited in the BOLD database under the name *A. acutifrons*. These sequences were included in the phylogenetic analysis of *Krüger* *et al. (2022)*, who suggested that there is more than one species included in *A. acutifrons*. The present study clarifies the taxonomic identity of the COI sequences and provides detailed morphological descriptions of the corresponding specimens. Because more than 80% of nominal *Ampharete* species remain unrepresented in genetic databases, the integration of molecular data with thorough morphological characterization is essential to generate a reliable molecular reference framework for Ampharetidae, and to enhance the accuracy of species identification.

The availability and good preservation of the Washington specimens, from which molecular sequences were obtained, enabled the detailed morphological comparisons and taxonomic reassessment presented in this study. In addition, the available photographs of living specimens allowed the description of pigmentation patterns, which can facilitate the identification of newly collected specimens. The above underscores the importance of

applying standardized protocols for the collection, handling and preservation of newly obtained material to maximize data preservation and facilitate future comparisons of living and preserved specimens as well as genetic information.

The examination of *A. cirrata* specimens from arctic-boreal localities revealed subtle morphological and molecular differences. These findings raise new questions about the genetic connectivity and reproductive isolation populations of *A. cirrata* and presents opportunities for future research.

The results of this study show that the identification of *Ampharete* can be challenging, especially when dealing with juvenile specimens. As described in the Variation and Remarks sections of *A. paulayi* **n. sp.**, paratypes smaller than 15 mm in length have smooth tentacles and rounded cirri on abdominal neuropodia. Since no differences were observed in other diagnostic characters or in COI sequences, small paratypes were considered to have reached full segmental development of all 29 body segments as in adult specimens (*e.g.*, holotype, 33 mm in length), but had not completed the development of the buccal tentacles and abdominal pinnules. Therefore, I hypothesize that the development of the tentacular surface and neuropodial cirri in *A. paulayi* **n. sp.**, and probably in other *Ampharete* species, occurs in late juvenile stages.

The use of Methyl green allowed the recognition of a distinctive staining pattern in each of the *Ampharete* species included in this study. Methyl green dye use has revealed interspecific differences in ampharetids (*e.g.*, *Jirkov, 2011*; *Kim, Kim & Jeong, 2025*) and other tubicolous annelids (*e.g.*, *Winsnes, 1985*; *Tovar-Hernández, de León-González & Bybee, 2017*), due to its affinity to glandular areas. Because it is a temporary stain and disappears in alcohol after few hours (*Winsnes, 1985*), Methyl green can be used on type material without damaging it. Therefore, the description of Methyl green staining patterns should become standard practice in future descriptions of *Ampharete* and other ampharetid species.

## CONCLUSIONS

The detailed morphological study of specimens previously recorded as *A. acutifrons* from Washington allow the identification three species, including the new species *A. paulayi* **n. sp**. Based on morphological and COI sequences data, a hypothesis on the development of buccal tentacles and abdominal neuropodia in *Ampharete* species is proposed. The findings of this study raise new questions about the morphological variation and geographic distribution of *A. cirrata* for future research.

Given the complexity in the taxonomic identification of *Ampharete* species, largely due to the morphological similarities among species, this study highlights the value of Methyl green staining as a useful tool for distinguishing closely related species. The use of Methyl green dye is recommended as a standard practice in future descriptions of ampharetids.

## ACKNOWLEDGEMENTS

I thank Dr. Gustav Paulay (UF) for the loan of materials and permission to use his photographs of live specimens. Thanks to Dr. Emma Sherlock (BMNH), Dr. Lena Gustavsson (SMNH), Leslie Harris (LACM-AHF), and Dr. Karen Osborn (USNM) for

allow me to work with material from the collections in their care and for assisting me with sending material. Thanks are extended to Dr. Sergio I. Salazar-Vallejo (ECOSUR) and Dr. Jenna Moore (ZMH) for receiving material on my behalf and for their valuable comments to improve the manuscript. The critical reading and suggestion provided by Juan Moreira and two anonymous reviewers helped to improve the manuscript. This research is part of the PhD Thesis of Y. Chávez-López at ECOSUR.

### Funding

This work was supported by a PhD scholarship from Consejo Nacional de Humanidades, Ciencias y Tecnologías (CONAHCYT) No. 813893, and the following grants that made possible the study of the collection material: DAAD (Deutscher Akademischer Austauschdienst) Short-Term Grant (European collections), Kristian Fauchald Fellowship (Smithsonian NMNH), and the Polychaete Collection Study Grant (LACM-AHF). No additional external funding was received. The funders had no role in study design, data collection and analysis, decision to publish, or preparation of the manuscript.

### Grant Disclosures

The following grant information was disclosed by the authors:

PhD scholarship from Consejo Nacional de Humanidades, Ciencias y Tecnologías (CONAHCYT): 813893.

DAAD (Deutscher Akademischer Austauschdienst) Short-Term Grant (European collections).

Kristian Fauchald Fellowship (Smithsonian NMNH).

Polychaete Collection Study Grant (LACM-AHF).

### Competing Interests

The author declares that she has no competing interests.

### Author Contributions

- Yessica Chávez-López conceived and designed the experiments, performed the experiments, analyzed the data, prepared figures and/or tables, authored or reviewed drafts of the article, and approved the final draft.

### DNA Deposition

The following information was supplied regarding the deposition of DNA sequences:

The COI sequences used in this work are available at GenBank and BOLD: *Ampharete californica* (MT166986), *A. cirrata* voucher UF 8068 (as *A. acutifrons* BBPS506-19), *A. cirrata* voucher UF 8069 (as *A. acutifrons* BBPS507-19), *A. cirrata* voucher UF 8071 (as *A. acutifrons* BBPS508-19), *A. cirrata* voucher UF 8081 (as *A. acutifrons* BBPS509-19), *A. cirrata* (OM470635, OM470635), *A. falcata* (OR891674, OR891679, MG270098, MG270099), *A. finmarchica* (JX423738, OR891684, OR891672), *A. labrops* voucher UF 8202 (as *A. acutifrons* BBPS810-19), *A. labrops* (HM473291, HM473292, HM473290), A.

lindstroemi (OR891673, OR891681, OR891685), *A. paulayi* n. sp. voucher UF 7977
(as *A. acutifrons* BBPS043-19), *A. paulayi* n. sp. voucher UF 7937 (as *A. acutifrons*
BBPS1016-19), *A. paulayi* n. sp. voucher UF 7692 (as *A. acutifrons* BBPS1017-19),
*A. santillani* (OR891683, OR891670, MG230531, MG230532), *A. undecima* (OR891675,
OR891676, OR891678), *Anobothrus gracilis* (MG270106, JX423739), *Sabellides manriquei*
(CMBIA290-11), *S. octocirrata* (JX423770, OR891680, OR891671).

## Data Availability

The raw data is available in the Supplemental File.

## New Species Registration

The following information was supplied regarding the registration of a newly described
species:

Publication LSID: urn:lsid:zoobank.org:pub:F771A2B4-4F5F-4168-BBAD-
A519175C83AE

*Ampharete paulayi* n. sp. LSID: urn:lsid:zoobank.org:act:DFDDEF26-C963-412D-
93B0-98177A08EB42.

## Supplemental Information

Supplemental information for this article can be found online at http://dx.doi.org/10.7717/
peerj.20457#supplemental-information.

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
