# Peer review of "A new species of Ampharete Malmgren, 1866 (Annelida: Ampharetidae) from Washington and redescription of A. cirrata Webster & Benedict, 1887 and A. labrops Hartman, 1961"

_PeerJ, doi:10.7717/peerj.20457_

## Round 0.1 · original submission · Minor Revisions

· Academic Editor

Minor Revisions

Dear Dr. Chávez-López, I kindly request you to make the minor corrections recommended by the reviewers and hope that the new version of this article will be approved for publication.

·

Basic reporting

No comments

Experimental design

No comments

Validity of the findings

No comments

Additional comments

This is a fine taxonomic work that describes a new species and redescribed to related species, after examination of a fair number of specimens. All methods are well described.The descriptions are detailed including good micrographs of relevant body taxonomic characters; the paper also includes phylogenetic molecular analyses based on available information. The discussion of the species are well articulated and the author provides plausible hypotheses for development of tentacles and parapodia features comparing large and small specimens. In all, this is how polychaete taxonomy should be done nowadays.
I am only miss some line drawings but it seems that this is not a common practice anymore; anyway, micrographs are clear and well iluminated.
I have just included some minor corrections and a few comments in the attached version of the manuscript and table 1. The figure legends are missing for the original files or rather I could not find them.

Reviewer 2 ·

Basic reporting

The paper is well written. Th background, methods and results are presented clearly.

Experimental design

The methods used are appropriate. I only have two small points that may need clarification in the paper:

Line 102, This paragraph says that all sequences were previously published in BOLD, while in the table, several sequences are referenced as "this study". This is a bit confusing. Does that mean that these sequences were produced and made public outside a particular project, and not specifically for this study? Or were they produced for this study, in which case a section of amterial and methods is missing.

Figure 1: No mention of bootstrap or other support is made in this text, and none is shown on the tree. This is a bit unconventional. On such a small dataset, estimating support values would not be particuarly time consuming. Please add them if you have them.

Validity of the findings

The new species is well supported by both molecular and morphological data. The descriptions are thorough and very well illustrated. Particular emphasis has been put on describing and clarifying the terminology (e.g., regarding the neuropodial cirri) or the way structures are counted/numbered (e.g., segments or branchiae), which is very helpful.

Additional comments

Once comment I could make is that the new species is presented first. If the revision and redesctiption of the other two species is necessary in order to be able to describe the new species, it could make sense that they appear first.

Reviewer 3 ·

Basic reporting

In this paper the author has reevaluated the identity of publicly available COI sequences, initially identified as Ampharete acutifrons, from Washington, USA. The author has reexamined the sequenced specimens and, based on morphology, concluded that the specimens belonged to two known species and one species new to science. The author has described the new species and, based on type material, redescribed the other species. The descriptions are detailed and well illustrated with detailed photos.

The paper is well written, with good English. There are, however, some improvements which need to be made to the text. The author has explained some of the terminology used in the descriptions in the Method-section, but not all, which can be confusing for someone not familiar with ampharetids. In particular, the complexity of the prostomium needs to be better explained and “intermediate uncinigers”, pinnulae and tori needs to be defined. Also, the author has a paragraph about cirri/papillae in the Result-section. This is better suited in methods and should be rewritten to clarify the difference between names of characters versus state of the characters, because as she states, this is confusing. Cirri is by definition long and thin, “dorsal cirri” on the other hand can be long, short or absent. Same goes for “lateral cirri” and “nephridial papillae”. The anus on the species described in this paper is surrounded by cirri, but in other Ampharete species the anus might be surrounded by papillae. For instance, Holthe (1986) in Terebellomorpha referred to this character as “anal papillae” and cirriform anal papillae when they were long. Care should be taken throughout the text to make sure that wherever cirri or papillae are used it is clear whether it is a character or a structure in that form. The author is also inconsistent when it comes to the use of paleae and paleal chaeta. Each specimen only has one pair of paleae, but each paleae has multiple chaetae. There are multiple places in the manuscript where paleae should be replaced with paleal chaetae.

Experimental design

no comment

Validity of the findings

no comment

Annotated reviews are not available for download in order to protect the identity of reviewers who chose to remain anonymous.

---

## Round 0.2 · accepted · Accept

· Academic Editor

Accept

Dear Dr. Chávez-López, I congratulate you on the acceptance of this article for publication.

·

Basic reporting

No comment

Experimental design

No comment

Validity of the findings

No comment

Additional comments

The new version of the manuscript succesfully addreses the comments and suggestions raised on the previous version. I have just found a few minor mistakes that I highlighted in the attached pdf version.

Reviewer 2 ·

Basic reporting

no comment

Experimental design

no comment

Validity of the findings

no comment

Additional comments

no comment

Reviewer 3 ·

Basic reporting

no comment

Experimental design

no comment

Validity of the findings

no comment

Additional comments

The author has responded to all my previous comments. I have nothing new to add. It is a very nice paper.